# MoGe-2: Accurate Monocular Geometry with Metric Scale and Sharp Details

**Ruicheng Wang**[1,2*] **Sicheng Xu**[2] **Yue Dong**[2] **Yu Deng**[2] **Jianfeng Xiang**[3,2*] **Zelong Lv**[1,2*]
**Guangzhong Sun**[1] **Xin Tong**[2] **Jiaolong Yang**[2†]
[1]USTC  [2]Microsoft Research  [3]Tsinghua University

## Abstract

We propose MoGe-2, an advanced open-domain geometry estimation model that recovers a metric scale 3D point map of a scene from a single image. Our method builds upon the recent monocular geometry estimation approach, MoGe [61], which predicts affine-invariant point maps with unknown scales. We explore effective strategies to extend MoGe for metric geometry prediction without compromising the relative geometry accuracy provided by the affine-invariant point representation. Additionally, we discover that noise and errors in real data diminish fine-grained detail in the predicted geometry. We address this by developing a unified data refinement approach that filters and completes real data from different sources using sharp synthetic labels, significantly enhancing the granularity of the reconstructed geometry while maintaining the overall accuracy. We train our model on a large corpus of mixed datasets and conducted comprehensive evaluations, demonstrating its superior performance in achieving accurate relative geometry, precise metric scale, and fine-grained detail recovery – capabilities that no previous methods have simultaneously achieved.

## 1 Introduction

Estimating 3D geometry from a single monocular image is a challenging task with numerous applications in computer vision and beyond. Recent advancements in Monocular Depth Estimation (MDE) and Monocular Geometry Estimation (MGE) have been driven by foundation models trained on large-scale datasets [67, 68, 44, 27, 61, 7]. Compared to depth estimation, MGE approaches often also predict camera intrinsics, allowing pixels to be lifted into 3D space, thus enabling a broader range of applications.

Despite the promising results of recent MGE models, they remain far from perfect and broadly applicable. We expect an ideal MGE method to excel in three key areas: *1) geometry accuracy*, *2) metric prediction*, and *3) geometry granularity*. While accurate global and relative geometry is essential, metric scale is crucial for real-world applications such as SLAM [54, 36], Autonomous Driving [56, 77], and Embodied AI [82, 81, 46]. In addition, recovering fine-grained details and sharp features is also critical for these fields as well as others like image editing and generation [79, 75, 60]. To our knowledge, no existing method addresses all these needs well simultaneously.

In this paper, we introduce a new MGE method towards achieving these goals, while maintaining a simple, principled, and pragmatic design. Our method is built upon the recent MoGe approach [61], which predicts affine-invariant point maps from single images and achieves state-of-the-art geometry accuracy. The cornerstone of MoGe is its optimized training scheme, including a robust and optimal point cloud alignment solver as well as a multi-scale supervision method which enhances local

---

*Work done during internship at Microsoft Research

†Corresponding author

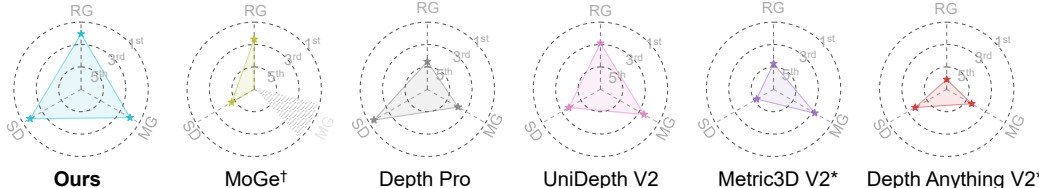

Figure 1: Rankings in comprehensive evaluations. Our method achieves accurate **R**elative **G**eometry (RG), precise **M**etric **G**eometry (MG), and **S**harp **D**etail recovery (SD) - capabilities not simultaneously achieved by previous approaches. ∗ Methods do not predict camera intrinsics and are evaluated on depth only. † MoGe [61] does not predict metric scale. Please refer to Sec. 4.1 for details.

geometry accuracy. Our work extends MoGe [61] by introducing metric geometry prediction capabilities and improving its geometry granularity to capture intricate details.

For metric geometry estimation, a straightforward solution involves directly predicting absolute point maps in metric space. However, this is suboptimal due to the focal-distance ambiguity issue [61]. To address this, we explore two simple, intuitive, yet effective alternatives. The first uses a shift-invariant point map representation which directly integrates metric scale into point map prediction. The second retains affine-invariant representation but additionally predicts a global scale factor in a decoupled manner. Both strategies mitigate the focal-distance ambiguity, but the latter yields more accurate results, likely due to its well-normalized point map space that better preserves relative geometry.

In the latter regard, we propose a pragmatic data refinement approach to generate sharp depth labels for real-world training data. Real data labels are often noisy and incomplete, particularly at object boundaries, which impede fine geometry detail learning. Previous works such as Depth Anything V2 [68] have opted to use only synthetic data labels, sacrificing the geometry accuracy, despite being sharp upon 2D visualization. Similarly, Depth Pro [7] employs only synthetic data in their second of the two stages. In contrast, we embrace real data throughout the training to ensure high geometry accuracy – a critical goal for our method. Our pipeline filters mismatched or false depth values in real data, primarily found around object boundaries, followed by edge-preserving depth inpainting to fill missing regions using a model trained on synthetic data. This approach results in significantly finer details, with geometry accuracy comparable to models trained on full unprocessed real data.

We train our model on an extensive collection of synthetic and real datasets and conduct a comprehensive evaluation across various datasets and metrics. Experiments demonstrate that our method achieves superior performance in terms of relative geometry accuracy, metric scale precision, and fine-grained detail recovery, surpassing multiple recently proposed baselines, as shown in Fig. 1.

**Our contributions are summarized as follows:**

- We introduce a Metric MGE framework with the representation of decoupled affine-invariant pointmap and global scale. We provide both insights and empirical evidences for this design.

- We propose a pragmatic real data refinement approach which enables sharp detail prediction while maintaining the generality by fully leveraging large scale real data.

- Our method achieves state-of-the-art results in both geometry accuracy and sharpness, significantly surpassing prior methods in global and local geometry accuracy.

We believe our method enhances monocular geometry estimation's potential in real-world applications and can serve as a foundational tool facilitating diverse tasks such as 3D world modeling, autonomous systems, and 3D content creation.

## 2 Related Works

**Monocular metric depth estimation.** Early works in this field [13, 15, 71, 4, 20] primarily focused on predicting metric depth in specific domains like indoor environments or street views, using limited data from certain RGBD cameras or LiDAR sensors. With the increasing availability of depth data from various sources, recent methods [5, 74, 23, 67, 68, 44, 7] have aimed to predict metric depth in open-domain settings. For example, Metric3D [74, 23] utilized numerous metric depth datasets

and introduced a canonical camera transformation module to address metric ambiguity from diverse data sources. ZoeDepth [5] built on a relative depth estimation framework [47, 6] that is pre-trained on extensive non-metric depth data and employed domain-specific metric heads. UniDepth [44, 45] instead simultaneously learned from metric and non-metric depth data to improve generalizability. Our method focuses on metric geometry estimation and also enables metric depth estimation by directly using the z-channel from the predicted point map, outperforming existing approaches in open-domain metric depth predictions.

**Monocular geometry estimation.**    This task aims to predict the 3D point map of a scene from a single image. Common approaches [72, 73, 44, 45] decouple point map prediction into depth estimation and camera parameter recovery. For instance, LeRes [72] estimates an affine-invariant depth map and camera focal and shift with two separate modules. UniDepth series [44, 45] predicted camera embeddings and facilitate depth map prediction with the estimated camera information. Along another line, DUSt3R [62] proposed an end-to-end 3D point map prediction framework for stereo images, bypassing explicit camera prediction. In a similar vein, MoGe [61] predicted an affine-invariant point map for monocular input, achieving state-of-the-art performance with a robust and optimal alignment solver. However, it does not account for metric scale and lacks the finer details, thereby limiting its applicability in many downstream tasks.

**Depth prediction with fine-grained details.**    Numerous methods [40, 35, 45, 68, 27, 7, 25, 39, 80, 66] have been developed to recover fine-grained details in depth prediction. Some [40, 35] enhance local details by fusing depth maps for image patches, but suffer from stitching artifacts. Other works [27, 16, 18] leverage pretrained image diffusion models [50] to generate detailed depth maps. Depth Anything V2 [68] highlights the importance of synthetic data labels by finetuning a DINOv2 [43] encoder with synthetic data and distilling from a larger teacher model. However, synthetic-to-real domain gaps persist and hinder the prediction accuracy. Depth Pro [7] integrates multi-patch vision transformers [11] and a synthetic data training stage, significantly improving depth map sharpness over previous methods, but still falls short in geometric accuracy. In contrast, our model achieves both fine detail recovery and precise geometry through the joint use of synthetic data and real data with a carefully designed real data refinement strategy.

**RGB-depth data misalignment artifacts**    Despite their overall accuracy, depth datasets captured with LiDAR [64, 53, 56, 19] or structure-from-motion (SfM) reconstructions [78, 70, 34] often exhibit various misalignment artifacts. Common issues include spatial misalignment caused by sensor asynchrony [76], ghost surfaces, and incomplete surface reconstruction [70]. Existing methods address LiDAR-specific issues using stereo cues [56] or epipolar geometry [84], while SfM artifacts are mitigated by regenerating depth maps with neural rendering [37, 2]. However, these approaches are often tailored to specific types of artifacts or rely on computationally expensive pipelines. We propose a unified data refinement approach that can handle diverse misalignment artifacts in RGB-depth data regardless of their source or underlying error patterns.

## 3   Methodology

Our method processes a single image to predict the 3D point map of the scene, achieving accurate relative geometry, metric scale, and fine-grained detail. It builds upon the recent MoGe approach [61] that focuses on affine-invariant point map reconstruction (Sec. 3.1). We explore effective strategies to extend it to accurate metric geometry estimation (Sec. 3.2). Additionally, we develop a data refinement approach that fully leverages real-world training data to achieve both precise and detailed geometry reconstruction simultaneously (Sec. 3.3).

### 3.1   Preliminaries: MoGe

Given a single image $I \in \mathbb{R}^{H \times W \times 3}$, MoGe estimates an affine-invariant 3D point map $\hat{P} \in \mathbb{R}^{H \times W \times 3}$ with an unknown global scale and shift relative to the ground truth geometry $P$, achieved by learning through a robust $L_1$ loss:

$$\mathcal{L}_G = \sum_{i \in \mathcal{M}} \frac{1}{z_i} \left\| s^* \hat{p}_i + t^* - p_i \right\|_1, \tag{1}$$

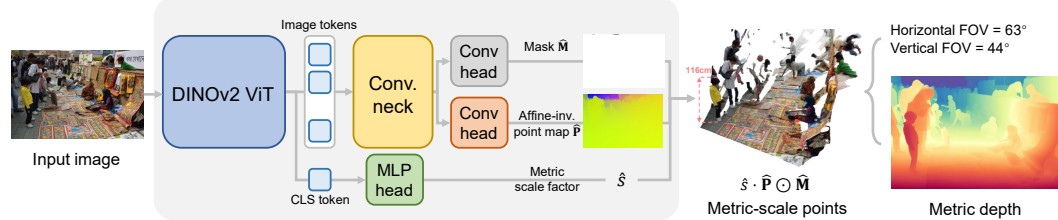

Figure 2: Overview of our model architecture. With the key insight of decoupling metric MGE into affine-invariant point map prediction [61] and global scale recovery, our network design extends MoGe [61] with an additional head for metric scale prediction. This design preserves the benefits of affine-invariant representations for accurate relative geometry while enabling metric scale estimation with the global features captured by the ViT encoder's classification token.

where $\mathcal{M}$ is the valid mask of ground truth point map, $1/z_i$ is a weighting scalar using inverse ground truth depth, and $s^*$ and $t^*$ are the optimal global scale and shift alignment factors derived by *a robust and optimal (ROE) alignment solver* [61],

$$(s^*, t^*) = \mathrm{argmin}_{s,t} \sum_{i \in \mathcal{M}} \frac{1}{z_i} \left\| s\hat{p}_i + t - p_i \right\|_1 . \tag{2}$$

To enhance local geometry accuracy, it further applies the robust supervision in Eq. (1) to multi-scale local spherical regions

$$\mathcal{S}_j = \{ i \mid \| p_i - p_j \| \le r_j, i \in \mathcal{M} \}, \tag{3}$$

centered at sampled ground truth point $p_j$ with different radius $r_j$. After obtaining the affine-invariant point map, the camera's focal and shift can be recovered by a simple and efficient optimization process (see [61] for more details).

While MoGe accurately predicts relative geometries, it falls short in addressing metric scale and lacks fine-grained details, limiting its broader applications. We explore these challenges and propose effective solutions to achieve accurate metric scale geometry estimation and fine-grained detail reconstruction, as detailed below.

## 3.2 Metric Scale Geometry Estimation

We explore two alternatives to extend MoGe with metric scale prediction, with corresponding design choices illustrated in Fig. 3.

**Shift-invariant geometry prediction.** As illustrated in Fig. 3-1, a natural extension of MoGe is to predict a shift-invariant point map by absorbing the metric scale $s$ into the affine point map, while computing only the global shift $t$ via ROE alignment during training and re-solving it again at inference time. This design bypasses the focal-distance ambiguity [61] and yields reasonable metric reconstruction results (Tab. 4).

However, due to the large variation in scene scale across open-domain images (*e.g.*, indoors vs. landscapes), the predicted values in shift-invariant space span a wide range. This makes scale learning less stable, and inaccurate scale predictions can produce large gradients that interfere with relative geometry learning (*i.e.*, the middle section of Tab. 4). This motivates our choice to decouple scale estimation from the point map prediction entirely.

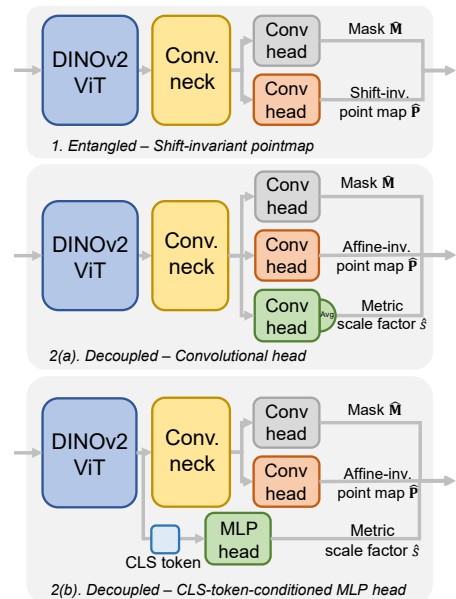

Figure 3: Model design choices for metric scale geometry estimation.

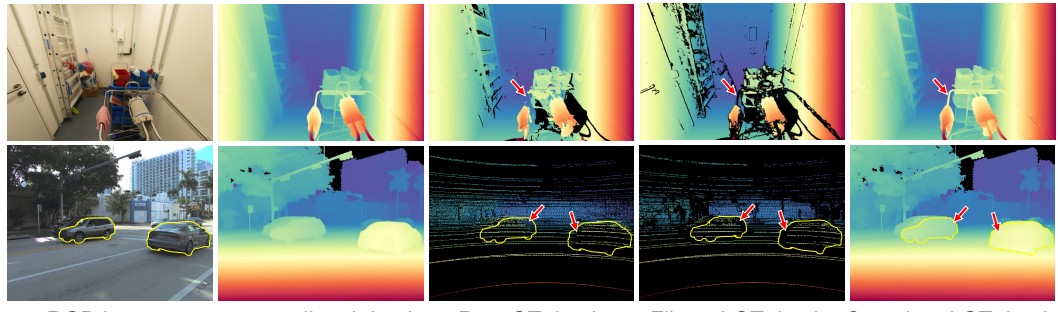

| RGB Image | $G_{\text{syn}}$ predicted depth | Raw GT depth | Filtered GT depth | Completed GT depth |

Figure 4: Filtering and completion for real captured datasets. **Top:** The ScanNet++ dataset [70], based on SfM reconstruction, struggles with thin structures and metallic surfaces. Our filtering process removes these artifacts, and our completion scheme reconstructs depth maps that maintain robust absolute depth while compensating for local details that align with the image. **Bottom:** In the Argoverse2 dataset [64], depth and color image discrepancies occur due to temporally unsynchronized sensors. Marking the vehicle boundary in color images (yellow lines) indicates a significant mismatch.

**Scale and relative geometry decomposition.** To prevent scale affecting relative geometry accuracy, we maintain the geometry branch for affine-invariant point map as in MoGe, and introduce an additional branch for scale prediction with exclusive supervision:

$$\mathcal{L}_s = \|\log(\hat{s}) - \text{stopgrad}(\log(s^*))\|_2^2, \tag{4}$$

where $\log(\hat{s})$ is the predicted metric scale in logarithmic space, and $s^*$ is the optimal scale calculated *online* by Eq. (2) between the predicted affine-invariant point map and the ground truth using the ROE solver. The final metric scale geometry is obtained by multiplying the predicted scale with the affine-invariant point map. We explore two design options for the additional scale prediction branch:

*(a) Convolutional head.* A naive design, as shown in Fig. 3-2(a), is to add a convolution head to output a single scale value, sharing the convolution neck with the affine-invariant point map. However, this approach does not improve relative geometry and worsens metric scale predictions (see Tab. 4). We suspect that simply adding a convolution head results in most information being processed in the convolution neck, which fails to decouple scale prediction from its effect on relative geometry. Moreover, the small output head is ineffective at aggregating local features from the convolution neck, while accurate metric scale prediction requires global information.

*(b) CLS-token-conditioned MLP.* To better decouple relative geometry and metric scale predictions, our second design (Fig. 3-2(b)) uses an MLP head to learn the metric scale directly from the DINOv2 encoder's classification (CLS) token (see Fig. 2). The global information in the token enables the network to predict an accurate metric scale. As demonstrated in Table 4, such simple design improves metric geometry accuracy compared to the convolution head method while maintaining accurate relative geometry. Thus, we adopt this design as our final configuration.

### 3.3 Real Data Refinement for Detail Recovery

We found that the MoGe model struggles to accurately reconstruct fine-grained structures due to noise and incompleteness in real training data. Previous studies [68, 27] have also noted this issue and suggest training with synthetic data of sharp labels and pretrained vision foundation models for real-world generalization. However, this still limits geometry accuracy because synthetic data rarely captures real-world diversity. Therefore, using real datasets while reducing their noise and incompleteness is crucial for accurate geometry estimation. To address this, we design a real data refinement pipeline that incorporates synthetic labels to mitigate common failure patterns in real data.

**Failure pattern analysis.** Real data often originated from LiDAR scans or Structure from Motion (SfM) reconstructions. LiDAR data can suffer from synchronization issues, causing depth and color mismatches, especially at object boundaries. SfM data might miss structures like reflective surfaces, thin structures, and sharp boundaries, as shown in Fig. 4. Our refinement approach leverages the fact that models trained on synthetic data achieve exact color-depth matching and capture sharp, complete

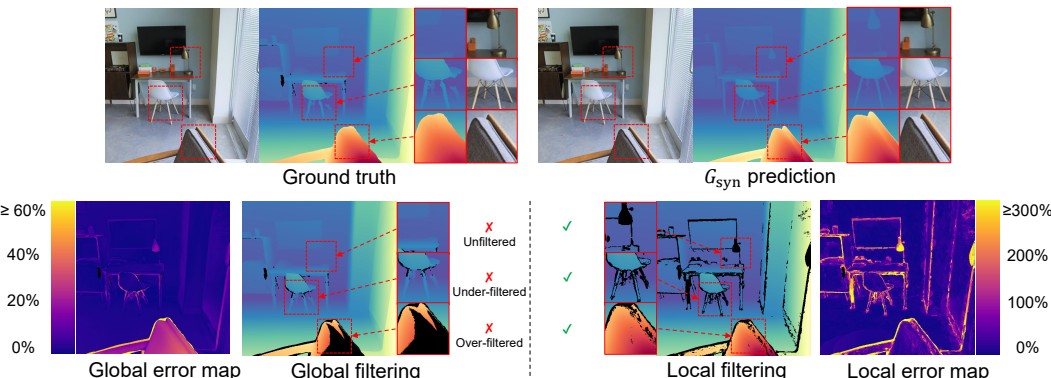

Figure 5: Our mismatch filtering scheme with local geometry alignment effectively avoids depth bias of the predicted results and helps to identify correct artifacts in the real data, whereas a global alignment fails to address the bias and introduces foreground errors, making it unsuitable for filtering.

local geometries. These pseudo labels can help filter incorrect depths and fill in missing parts in real data given accurate local geometries.

**Mismatch filtering.** To filter real captured depth data, we train a MoGe model solely on synthetic data, denoted as $G_{\text{syn}}$. The model is then applied to real images to infer geometry, which serves as a reference for filtering. While the predicted local structures are generally plausible, $G_{\text{syn}}$ often yields biased estimates of the overall scene geometry and layout when used on real-world images. This bias arises from the lack of real-scene priors during training on purely synthetic data. As shown in Fig. 5, such bias can result in incorrect filtering when global errors are considered. Therefore, we focus on comparing the local structures between the real and predicted point maps.

Specifically, given the real-captured points $\{\boldsymbol{p_i}\}$, corresponding predictions $\{\hat{\boldsymbol{p}}_i\}$ by $G_{\text{syn}}$, and the mask $\mathcal{M}$ of valid ground truth, we select a spherical region $\hat{\mathcal{S}}_j$ centered at each estimated point $\hat{\boldsymbol{p}}_j$ with a specific radius $\hat{r}_j$:

$$\hat{\mathcal{S}}_j = \left\{ i \mid \|\hat{\boldsymbol{p}}_i - \hat{\boldsymbol{p}}_j\| \leq \hat{r}_j, i \in \mathcal{M} \right\}. \tag{5}$$

Within this local region $\hat{\mathcal{S}}_j$, we align the corresponding real-captured points $\{\boldsymbol{p}_i\}_{i \in \hat{\mathcal{S}}_j}$ with the predictions $\{\hat{\boldsymbol{p}}_i\}_{i \in \hat{\mathcal{S}}_j}$ via the ROE solver and mark a real-captured point as an outlier if deviates from the predictions beyond the specified radius, forming a set $\mathcal{O}_j$:

$$\mathcal{O}_j = \left\{ i \mid \|s_j^* \boldsymbol{p}_i + \boldsymbol{t}_j^* - \hat{\boldsymbol{p}}_i\| > \hat{r}_j, i \in \hat{\mathcal{S}}_j \right\}, \tag{6}$$

with $(s_j^*, \boldsymbol{t}_j^*)$ as optimal alignment factors for local regions. The outlier sets derived from all sampled local regions of different $\hat{r}_j$ are combined and excluded from the mask, yielding the final valid area

$$\mathcal{M}_{\text{filtered}} = \mathcal{M} \setminus \left( \bigcup_j \mathcal{O}_j \right). \tag{7}$$

Regarding the choice of radius $\hat{r}_j$ for each sampled anchor point, we follow the same multi-scale strategy in MoGe's local loss for capturing context at different spatial scales. Specifically, we set $\hat{r}_j = \alpha \cdot \hat{z}_j \cdot \frac{\sqrt{W^2 + H^2}}{2 \cdot \hat{f}}$, where $\hat{z}_j$ is the depth of $\hat{\boldsymbol{p}}_j$, $\hat{f}$ is the predicted focal length, $W$ and $H$ are image width and height, and $\alpha \in \{1/4, 1/16, 1/64\}$ controls the region's projected size approximately cover the respective ratio of the image size.

**Geometry completion.** After filtering out mismatch regions, we create a complete depth map by integrating the detailed structures predicted by $G_{\text{syn}}$ with the remaining ground truth depth. Specifically, we reconstruct the depth in the filtered-out regions $\{d_i^c\}_{i \in \mathcal{M}_{\text{filtered}}^c}$ using logarithmic-space Poisson completion:

$$\min \sum_{i \in \mathcal{M}_{\text{filtered}}^c} \|\nabla(\log d_i^c) - \nabla(\log \hat{d}_i)\|^2, \quad \text{s.t.} \quad d_i^c = d_i, \forall i \in \partial \mathcal{M}_{\text{filtered}}^c, \tag{8}$$

Table 1: Quantitative evaluation for *relative geometry*. The numbers are **averaged across the 10 evaluation datasets**. The metrics are visualized with a color gradient from green (best) to red (worst). Numbers in gray cells indicate that some test datasets were used in training. Non-applicable cases are marked with " - ". Detailed results on each dataset can be found in *suppl. materials*.

| Method | Point | | | | | | | | | Depth | | | | | | | | | Avg. |
| | Scale-inv. | | | Affine-inv. | | | Local | | | Scale-inv. | | | Affine-inv. | | | Affine-inv. (disp) | | | |
| | $Rel^p\downarrow$ | $\delta_1^p\uparrow$ | Rk.$\downarrow$ | $Rel^p\downarrow$ | $\delta_1^p\uparrow$ | Rk.$\downarrow$ | $Rel^p\downarrow$ | $\delta_1^p\uparrow$ | Rk.$\downarrow$ | $Rel^d\downarrow$ | $\delta_1^d\uparrow$ | Rk.$\downarrow$ | $Rel^d\downarrow$ | $\delta_1^d\uparrow$ | Rk.$\downarrow$ | $Rel^d\downarrow$ | $\delta_1^d\uparrow$ | Rk.$\downarrow$ | Rk.$\downarrow$ |
|---|---|---|---|---|---|---|---|---|---|---|---|---|---|---|---|---|---|---|---|
| ZoeDepth | - | - | - | - | - | - | - | - | - | 12.7 | 83.9 | 8.75 | 10.1 | 88.5 | 9.09 | 11.1 | 88.3 | 8.78 | 8.87 |
| DA V1 | - | - | - | - | - | - | - | - | - | 11.7 | 85.8 | 8.22 | 8.76 | 90.4 | 6.91 | 8.63 | 92.2 | 5.62 | 6.92 |
| DA V2 | - | - | - | - | - | - | - | - | - | 10.7 | 87.6 | 6.80 | 8.48 | 90.8 | 6.15 | 8.82 | 91.6 | 5.42 | 6.12 |
| Metric3D V2 | - | - | - | - | - | - | - | - | - | 7.92 | 91.8 | 3.39 | 7.66 | 92.9 | 4.53 | 9.51 | 89.4 | 6.17 | 4.70 |
| MASt3R | 14.5 | 82.1 | 5.45 | 11.6 | 86.0 | 5.45 | 8.09 | 92.2 | 5.40 | 11.2 | 86.5 | 7.65 | 9.38 | 89.1 | 7.97 | 11.6 | 87.8 | 8.60 | 6.75 |
| UniDepth V1 | 13.6 | 85.0 | 3.83 | 10.9 | 88.1 | 3.95 | 9.21 | 91.0 | 5.55 | 10.1 | 89.1 | 5.12 | 8.61 | 91.0 | 5.67 | 9.75 | 89.9 | 5.92 | 5.01 |
| UniDepth V2 | 11.6 | 87.7 | 2.98 | 8.56 | 91.9 | 2.55 | 6.34 | 94.9 | 3.10 | 8.61 | 90.8 | 3.10 | 6.42 | 93.9 | 2.80 | 7.35 | 93.0 | 2.75 | 2.88 |
| Depth Pro | 12.4 | 87.7 | 3.83 | 9.93 | 89.4 | 4.30 | 6.91 | 94.1 | 3.55 | 9.81 | 89.1 | 5.33 | 7.65 | 92.0 | 5.05 | 8.42 | 91.7 | 5.08 | 4.52 |
| MoGe | 7.46 | 94.1 | 2.14 | 5.69 | 95.2 | 2.14 | 5.50 | 95.6 | 2.05 | 5.77 | 94.5 | 2.72 | 4.51 | 96.1 | 2.94 | 5.58 | 95.4 | 3.17 | 2.53 |
| *Ours* | 10.8 | 88.5 | 2.40 | 7.98 | 91.7 | 2.23 | 5.33 | 95.9 | 1.35 | 7.35 | 92.2 | 2.12 | 5.62 | 94.8 | 2.02 | 6.66 | 93.8 | 2.17 | 2.05 |

Table 2: Quantitative evaluation for *metric geometry*. The numbers are **averaged across 7 datasets**.

| Method | Point | | | Depth | | | Depth (w/ GT Cam) | | | Avg. |
| | $Rel^p\downarrow$ | $\delta_1^p\uparrow$ | Rk.$\downarrow$ | $Rel^d\downarrow$ | $\delta_1^d\uparrow$ | Rk.$\downarrow$ | $Rel^p\downarrow$ | $\delta_1^p\uparrow$ | Rk.$\downarrow$ | Rk.$\downarrow$ |
|---|---|---|---|---|---|---|---|---|---|---|
| ZoeDepth | - | - | - | 39.3 | 49.9 | 5.90 | - | - | - | 5.90 |
| DA V1 | - | - | - | 31.8 | 54.8 | 5.50 | - | - | - | 5.50 |
| DA V2 | - | - | - | 29.9 | 56.6 | 4.43 | - | - | - | 4.43 |
| Metric3D V2 | - | - | - | - | - | - | 18.3 | 73.9 | 2.75 | 2.75 |
| MASt3R | 26.2 | 55.3 | 4.93 | 49.7 | 30.3 | 6.71 | - | - | - | 5.82 |
| UniDepth V1 | 12.1 | 87.2 | 2.71 | 23.2 | 67.5 | 3.32 | 21.4 | 68.6 | 2.50 | 2.84 |
| UniDepth V2 | 10.1 | 91.9 | 2.43 | 21.3 | 75.3 | 2.54 | 18.5 | 82.6 | 2.57 | 2.51 |
| Depth Pro | 13.7 | 81.9 | 3.29 | 27.6 | 54.4 | 4.36 | - | - | - | 3.83 |
| *Ours* | 8.19 | 93.6 | 1.64 | 15.7 | 76.8 | 2.21 | 13.6 | 87.4 | 2.00 | 1.95 |

Table 3: Evaluation of boundary sharpness using F1 scores ($\uparrow$) in percentages.

| Method | iBims-1 | HAMMER | Sintel | Spring | Avg. Rk.$\downarrow$ |
|---|---|---|---|---|---|
| ZoeDepth | 2.47 | 0.17 | 2.30 | 0.43 | 7.75 |
| DA V1 | 3.68 | 0.76 | 5.64 | 1.09 | 6.75 |
| DA V2 | 13.9 | 4.74 | 32.5 | 6.10 | 3.75 |
| Metric3D V2 | 7.36 | 1.40 | 25.3 | 7.23 | 5.25 |
| MASt3R | 1.24 | 0.05 | 1.72 | 0.15 | 9.50 |
| UniDepth V1 | 2.35 | 0.06 | 0.73 | 0.17 | 9.00 |
| UniDepth V2 | 11.2 | 4.40 | 39.7 | 7.08 | 3.75 |
| Depth Pro | 14.3 | 5.36 | 41.6 | 11.0 | 1.50 |
| MoGe | 11.4 | 3.89 | 26.3 | 8.36 | 4.67 |
| *Ours* | 17.9 | 5.42 | 35.2 | 8.63 | 1.75 |

where $\mathcal{M}_{\text{filtered}}^c$ is the complement area of $\mathcal{M}_{\text{filtered}}$, $\hat{d}_i$ and $d_i$ denote predicted depth by $G_{\text{syn}}$ and the real captured depth, respectively. This strategy ensures that the reconstructed depth aligns with the gradient of the predicted depth at local regions while maintaining the ground truth depth as the boundary condition.

Figure 4 illustrates our filtering and completion process. Our method effectively removes mismatched depths from LiDAR scans and fills in missing content in SfM-reconstructed depth maps. The completed depth map retains sharp geometric boundaries that align with the input image while preserving the robust absolute depth from the original map. The refined training data effectively enhances the model's sharpness and maintains accurate geometry estimation, as shown in Tab. 4.

## 4 Experiments

**Implementation details.** We train our model using a combination of 24 datasets with 16 synthetic datasets [10, 58, 49, 59, 42, 33, 14, 24, 83, 51, 63, 21, 1, 65, 55, 73], 3 LiDAR scanned datasets [17, 64, 53], and 5 SfM-reconstructed datasets [3, 78, 70, 34, 69]. We follow MoGe [61] to balance the weights and loss functions among different datasets, and also adopt their approach for image cropping and data augmentation. More details of the training datasets can be found in *suppl. material*.

We use DINOv2-ViT-Large as the backbone for the full model, and DINOv2-ViT-Base model for all ablation studies to ensure efficiency. Our convolutional heads follow MoGe's design but remove all normalization layers in order to significantly reduce inference latency. The models are trained with initial learning rates of $1 \times 10^{-5}$ for the ViT backbone and $1 \times 10^{-4}$ for the neck and heads. The learning rate decays by half every 25K steps. The full model is trained for 120K iterations with 32 NVIDIA A100 GPUs for 120 hours. Ablation models are trained for 100K iterations. Additional implementation details and runtime analysis are provided in the *supplementary material*.

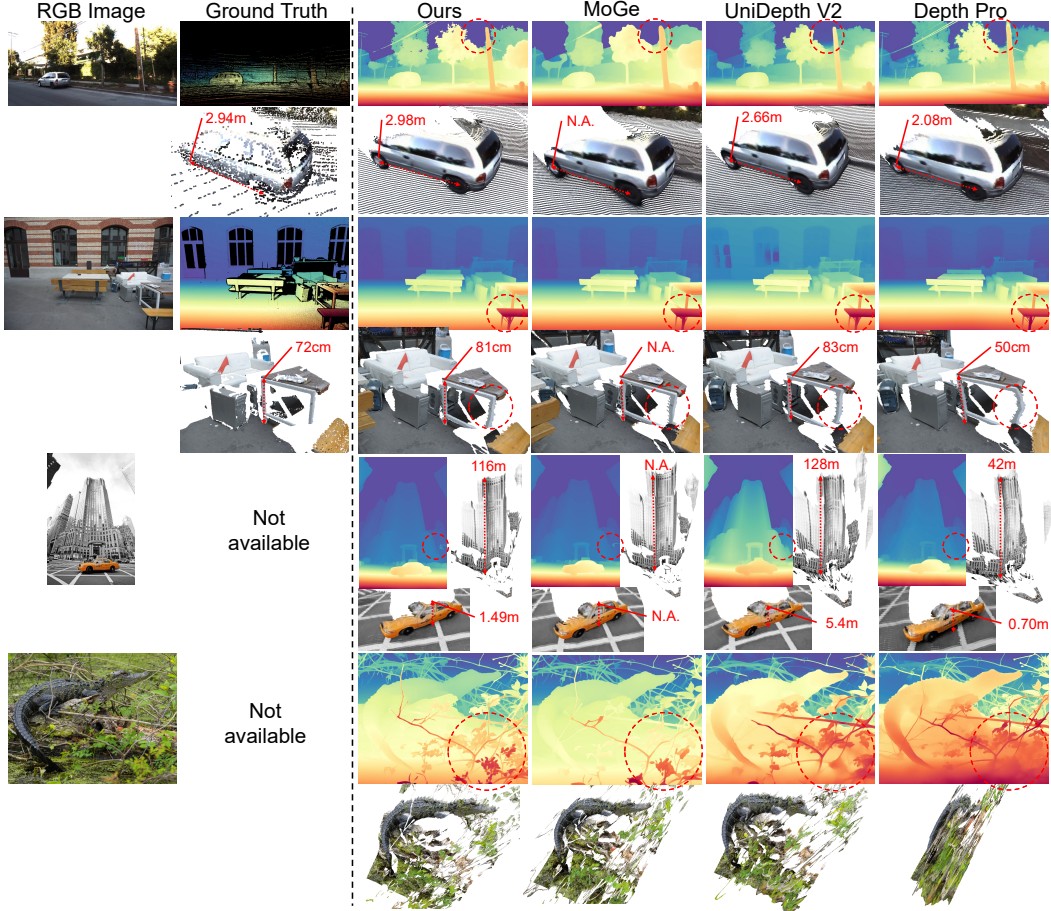

Figure 6: Qualitative comparison of metric scale point and disparity maps. The top two rows are selected from unseen metric scale test datasets. We also labeled key metric measurements in both the ground truth and the estimated geometry. Our estimated metric geometry best matches the ground truth and maintains sharp details. For open-domain inputs, our method produces reasonable geometry with rich details, while results of Depth Pro [7] are severely distorted. *Best viewed zoomed in.*

## 4.1 Quantitative Evaluation

**Benchmarks.** We evaluate the accuracy of our method on 10 datasets: NYUv2 [41], KITTI [56], ETH3D [52], iBims-1 [31, 30], GSO [12] , Sintel [8], DDAD [19], DIODE [57], Spring [38], and HAMMER [26]. These datasets encompass a wide range of domains, including indoor scenes, street views, object scans, and synthetic animations.

**Baselines.** We compare our method with several monocular geometry estimation methods, including UniDepth V1 and V2 [44, 45], Depth Pro [7], MoGe [61], MASt3R [32], as well as depth estimation baselines: Depth Anything V1 (DA V1) and V2 (DA V2) [67, 68], ZoeDepth [5] and Metric3D V2 [74]. We evaluate the performance of these methods based on relative scale geometry, metric scale geometry, and boundary sharpness.

**Relative geometry and depth.** While the primary goal of our method is to estimate metric scale geometry, measuring relative geometry provides valuable insights into how each method reconstructs the geometric shape from the input image. We employ the evaluation metrics of MoGe, measuring over the scale-invariant point maps, affine-invariant point maps, local point maps, scale-invariant depth, affine-invariant depth, and affine-invariant disparity.

Table 1 presents the average relative error – $\text{Rel}^p$ ($\|\hat{\boldsymbol{p}} - \boldsymbol{p}\|_2 / \|\boldsymbol{p}\|_2$) for point maps and $\text{Rel}^d$ ($|\hat{z} - z|/z$) for depth map), and the percentage of inliers ($\delta_1^p$, where $\|\hat{\boldsymbol{p}} - \boldsymbol{p}\|_2 / \|\boldsymbol{p}\|_2 < 0.25$, and $\delta_1^d$, where

Table 4: Ablation study results **averaged over 10 datasets**, conducted with a ViT-Base encoder.

| Configuration | Metric geometry | | | | Relative geometry | | | | | | | | | | | | Sharpness |
| | Point | | Depth | | Point | | | | | | Depth | | | | | | |
| | | | | | Scale-inv. | | Affine-inv. | | Local | | Scale-inv. | | Affine-inv. | | Affine-inv. (disp) | | |
| | $\text{Rel}^p\downarrow$ | $\delta_1^p\uparrow$ | $\text{Rel}^d\downarrow$ | $\delta_1^d\uparrow$ | $\text{Rel}^p\downarrow$ | $\delta_1^p\uparrow$ | $\text{Rel}^p\downarrow$ | $\delta_1^p\uparrow$ | $\text{Rel}^p\downarrow$ | $\delta_1^p\uparrow$ | $\text{Rel}^d\downarrow$ | $\delta_1^d\uparrow$ | $\text{Rel}^d\downarrow$ | $\delta_1^d\uparrow$ | $\text{Rel}^d\downarrow$ | $\delta_1^d\uparrow$ | F1 $\uparrow$ |
|---|---|---|---|---|---|---|---|---|---|---|---|---|---|---|---|---|---|
| Metric scale prediction design | | | | | | | | | | | | | | | | | |
| Entangled (SI-Log) | 10.0 | 90.7 | 17.9 | 68.6 | 12.9 | 86.2 | 10.3 | 88.8 | 8.21 | 93.0 | 9.83 | 89.0 | 7.97 | 92.0 | 9.03 | 91.1 | 10.7 |
| Entangled (Shift inv.) | 8.99 | 92.1 | 16.9 | 68.8 | 12.0 | 87.2 | 9.05 | 90.2 | 6.69 | 94.6 | 8.46 | 90.6 | 6.75 | 93.2 | 7.80 | 92.1 | 11.8 |
| Decoupled (Conv. head) | 9.62 | 91.4 | 17.7 | 68.4 | 12.2 | 86.3 | 9.15 | 90.0 | 6.34 | 94.9 | 8.46 | 90.2 | 6.62 | 93.2 | 7.74 | 92.1 | 12.7 |
| Decoupled (CLS-MLP) | 9.20 | 91.9 | 16.5 | 72.8 | 11.6 | 87.6 | 8.87 | 90.6 | 6.26 | 95.1 | 8.23 | 91.0 | 6.53 | 93.4 | 7.53 | 92.6 | 12.5 |
| Training data | | | | | | | | | | | | | | | | | |
| Synthetic data only | 12.4 | 87.3 | 21.7 | 65.0 | 12.3 | 85.9 | 9.77 | 88.9 | 6.42 | 94.9 | 9.04 | 89.6 | 7.25 | 92.5 | 8.37 | 91.6 | 13.3 |
| *w/* Raw real data | 9.01 | 92.2 | 15.8 | 75.7 | 11.4 | 87.8 | 8.69 | 90.7 | 6.37 | 94.9 | 8.40 | 90.4 | 6.63 | 93.3 | 7.69 | 92.2 | 10.3 |
| *w/* Improved real data | 9.20 | 91.9 | 16.5 | 72.8 | 11.6 | 87.6 | 8.87 | 90.6 | 6.26 | 95.1 | 8.23 | 91.0 | 6.53 | 93.4 | 7.53 | 92.6 | 12.5 |

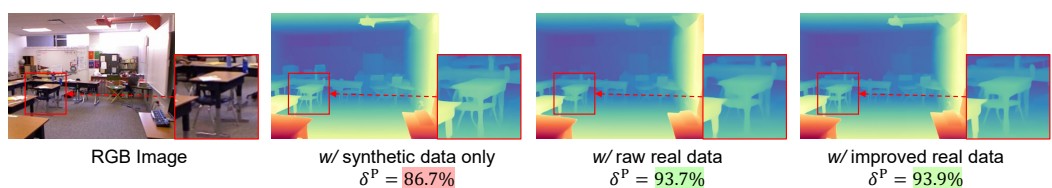

Figure 7: Showcase of ablation study on models trained with different data.

$\max(\hat{d}/d, d/\hat{d}) < 1.25)$ across the 10 test datasets, along with the average ranking among the 8 methods. Note that ZoeDepth, DA V1, DA V2, and Metric3D V2 are not evaluated for point settings due to the lack of camera focal prediction. Our method outperforms all existing baselines across every evaluation metric and achieves results comparable to the state-of-the-art relative geometry estimation method, MoGe. This demonstrates that our model *does not compromise the accuracy of relative geometry for achieving metric scale estimation.*

**Metric geometry and depth.** We evaluate the accuracy of metric scale geometry and depth using 7 datasets with metric scale annotations, including NYUv2 [41], KITTI [56], ETH3D [52], iBims-1 [31, 30], DDAD [19], DIODE [57] and HAMMER [26]. We measure the relative point error ($\text{Rel}^p$) and percentage of inliers ($\delta_1^p$) on estimated metric point maps. Similarly, we evaluate the metric depth accuracy via relative depth error ($\text{Rel}^d$) and depth inliers ($\delta_1^d$). Additionally, we evaluate metric depth estimation using ground truth camera intrinsics for methods that accept this input, which helps eliminate the influence of inaccuracies in the estimated camera intrinsics. As shown in Table 2, our method largely surpasses all existing methods across every metric measurement, demonstrating the advantages of our simple and effective design for decoupling metric scale and affine-invariant point estimation.

**Boundary sharpness.** To evaluate the sharpness of the estimated geometry, we use two synthetic datasets, Spring [38] and Sintel [8], as well as two real-world test datasets iBims-1 [31] and HAMMER [26], which contain high-quality, densely annotated geometry. We employ the boundary F1 score metric proposed by Depth Pro [7] to measure boundary sharpness. As shown in Table 3, our method achieves boundary sharpness comparable to that of Depth Pro [7] and significantly outperforms it in terms of both relative and metric scale geometry accuracy.

## 4.2 Qualitative Evaluation

Figure 6 presents a visual comparison of metric scale point maps and disparity maps estimated by different methods. We have annotated key metric scale measurements on both the ground truth and the estimated geometry to facilitate comparison of metric scale accuracy. Our method successfully produces metric scale geometry with sharp details, whereas MoGe and UniDepth V2 miss significant geometric details. Depth Pro exhibits reduced geometric accuracy, particularly in the open-domain test image of a crocodile.

### 4.3 Ablation Study

**Metric scale prediction.** In Sec. 3.2, we explored various strategies for accurate metric geometry estimation from open-domain images. We evaluate these configurations across the 10 test datasets using the aforementioned evaluation metrics. We also introduce a naive baseline that directly predicts a metric point map with entangled scale and shift factors using the commonly adopted SI-log loss [13].

Table 4 shows the evaluation results, highlighting the importance of a decoupled design that separates metric scale from relative geometry estimation to improve overall performance. For the scale prediction head, the MLP module outperforms the convolutional head, particularly in metric geometry. This indicates the importance of using global information to predict the metric scale and better decoupling of relative geometry from scale prediction.

**Real data refinement.** To evaluate the impact of our data refinement pipeline, we conducted ablation study using different data configurations for training – only synthetic data, raw real-world data, and our refined real-world data. As shown in Tab. 4, training exclusively on synthetic data yields the highest sharpness but significantly reduces geometric accuracy. This supports the effectiveness of using synthetic-data-trained model predictions to filter mismatched real data via local error. Training with real-world datasets enhances geometric accuracy but reduces sharpness. Our refined real-world datasets achieve nearly the same geometric accuracy as the original datasets while maintaining reasonable sharpness, as further confirmed by the visual comparison in Figure 7.

## 5   Conclusion

We have presented MoGe-2, a foundational model for monocular geometry estimation in open-domain images, extending the recent MoGe model to achieve metric scale estimation and fine-grained detail recovery. By decoupling the task into relative geometry recovery and global scale prediction, our method retains the advantages of affine-invariant representations while enabling accurate metric reconstruction. Alongside, we proposed a practical data refinement pipeline that enhances real data with synthetic labels, largely improving geometric granularity without compromising accuracy. MoGe-2 achieves superior performance in accurate geometry, precise metric scale and visual sharpness, advancing the applicability for monocular geometry estimation in real-world applications.

**Limitations.** Our method struggles with capturing extremely fine structures, such as thin lines and hair, and with maintaining straight and aligned structures under a significant scale difference between the foreground and background. The ambiguity in real-world metric scale can also lead to deviations in out-of-distribution scenarios. We aim to address these challenges by enhancing network architectures and incorporating more real-world priors in the future.

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

# A  Implementation Details

## A.1  Network architectures

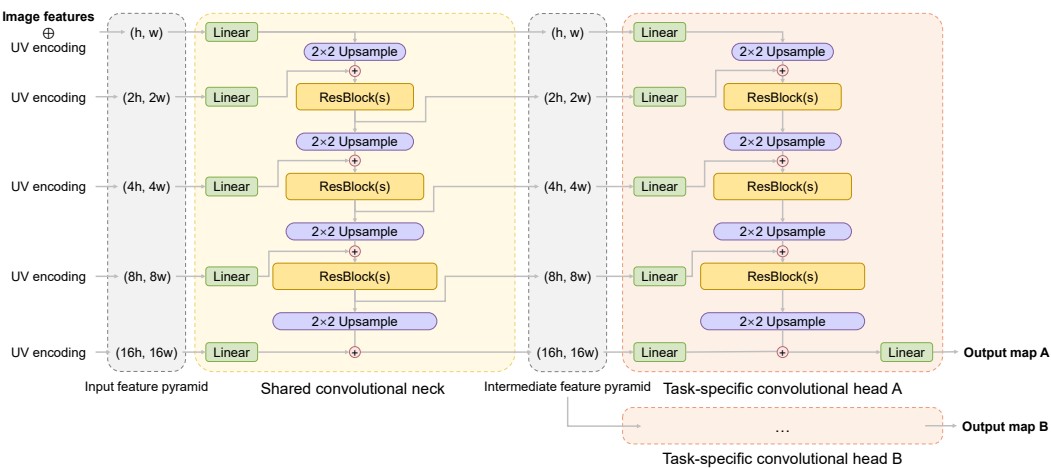

Figure A.1: Illustration of the convolutional neck and head module architectures.

The detailed architectures of our model components are described as follows.

**DINOv2 Image Encoder.**  Our model supports variable input resolutions by leveraging the interpolatable positional embeddings of DINOv2 [43]. The native resolution is determined by a user-specified number of image tokens. Given an input image of arbitrary size and a target number of tokens, we compute a patch-level resolution $h \times w$ that best matches the desired token count. The image is then resized to $(14h, 14w)$ to match DINOv2's input requirement, and encoded into $h \times w$ image tokens along with one classification token. We extract four intermediate feature layers from DINOv2—specifically, the 6th, 12th, 18th, and final transformer layers—project them to a common dimension, reshape their spatial size to $(h, w)$, and sum them to form the input for the dense prediction decoder.

**Convolutional Neck and Heads.**  Inspired by prior multi-task dense prediction architectures [48, 28, 61], we design a lightweight decoder consisting of a shared convolutional neck and multiple task-specific heads, as illustrated in Fig A.1. Both the neck and the heads are composed of progressive residual convolution blocks (ResBlocks) [22] interleaved with transpose convolution layers (kernel size 2, stride 2) for progressive upsampling from resolution $(h, w)$ to $(16h, 16w)$. Finally, the output map is resized through bilinear interpolation to match the raw image size. To reduce inference latency on modern GPUs, all normalization layers are simply removed from the ResBlocks, without affecting performance or training stability.

At each scale level of the neck, we inject a UV positional encoding, defined as a mapping of the image's rectangular domain into a unit circle, preserving the raw aspect ratio information. The resulting intermediate feature pyramid is shared across all heads, each of which independently decodes its respective output map. This design enables multi-scale feature sharing while maintaining head-specific decoding tailored to each prediction task.

**CLS-token-conditioned MLP Head.**  For scalar prediction, we use a two-layer MLP that takes the CLS token feature from DINOv2 as input and outputs a single scale factor, followed by an exponential mapping to ensure a positive scale output. The hidden layer size is equal to the input feature dimension.

## A.2  Training Data

The datasets used for training our model are listed in Tab. A.1. All datasets are publicly available for academic use, and their sampling weights follow the protocol established in MoGe [61].

Tab. A.2 provides a rough summary of the number of training frames used by several representative monocular geometry estimation methods. As there is no shared or standardized training set in this field, this table serves to contextualize the scale of training data across methods. Notably, model performance does not necessarily correlate with the amount of training data used.

Table A.1: List of datasets used to train our model.

| Name | Domain | #Frames | Type | Weight | Metric Scale |
|------|--------|---------|------|--------|--------------|
| A2D2[17] | Outdoor/Driving | 196K | LiDAR | 0.8% | ✓ |
| Argoverse2[64] | Outdoor/Driving | 1.1M | LiDAR | 7.1% | ✓ |
| ARKitScenes[3] | Indoor | 449K | SfM | 8.3% | ✓ |
| BlendedMVS[69] | In-the-wild | 115K | SfM | 11.5% | |
| MegaDepth[34] | Outdoor/In-the-wild | 92K | SfM | 5.4% | |
| ScanNet++[9] | Indoor | 176K | SfM | 4.6% | ✓ |
| Taskonomy[78] | Indoor | 3.6M | SfM | 14.1% | ✓ |
| Waymo[53] | Outdoor/Driving | 788K | LiDAR | 6.2% | ✓ |
| ApolloSynthetic[1] | Outdoor/Driving | 194K | Synthetic | 3.8% | ✓ |
| EDEN[73] | Outdoor/Garden | 369K | Synthetic | 1.2% | |
| GTA-SfM[58] | Outdoor/In-the-wild | 19K | Synthetic | 2.7% | ✓ |
| Hypersim[49] | Indoor | 75K | Synthetic | 4.8% | ✓ |
| IRS[59] | Indoor | 101K | Synthetic | 5.4% | ✓ |
| KenBurns[42] | In-the-wild | 76K | Synthetic | 1.5 % | |
| MatrixCity[33] | Outdoor/Driving | 390K | Synthetic | 1.3% | ✓ |
| MidAir[14] | Outdoor/In-the-wild | 423K | Synthetic | 3.8% | ✓ |
| MVS-Synth[24] | Outdoor/Driving | 12K | Synthetic | 1.2% | ✓ |
| Structured3D[83] | Indoor | 77K | Synthetic | 4.6% | ✓ |
| Synthia[51] | Outdoor/Driving | 96K | Synthetic | 1.1% | ✓ |
| Synscapes[65] | Outdoor/Driving | 25K | Synthetic | 1.9% | ✓ |
| UnrealStereo4K [55] | In-the-wild | 8K | Synthetic | 1.6 | ✓ |
| TartanAir[63] | In-the-wild | 306K | Synthetic | 4.8% | ✓ |
| UrbanSyn[21] | Outdoor/Driving | 7K | Synthetic | 2.0% | ✓ |
| ObjaverseV1[10] | Object | 167K | Synthetic | 4.6% | |

Table A.2: Summary of labeled training frame counts and pretrained backbones for the models compared in this paper.

| Method | #Total Training Frames | Pretrained Backbone |
|--------|------------------------|---------------------|
| ZoeDepth [5] | $\sim$ 2M | MiDaS BEiT384-L [47] |
| DA V1 [67] | 1.5M (+ 62M pseudo-labeled) | DINOv2 ViT-Large |
| DA V2 [68] | 595K (+ 62M pseudo-labeled) | DINOv2 ViT-Large |
| Metric3D V2 [23] | 16M | DINOv2 ViT-Large |
| UniDepth V1 [44] | 3.7M | DINOv2 ViT-Large |
| UniDepth V2 [45] | 16M | DINOv2 ViT-Large |
| Depth Pro [7] | $\sim$ 6M | DINOv2 ViT-Large |
| MoGe [61] | 9M | DINOv2 ViT-Large |
| *Ours* | 8.9M | DINOv2 ViT-Large |

## A.3 Evaluation Protocol

**Relative Geometry** We follow the evaluation protocol of alignment in MoGe [61]. Predictions and ground truth are aligned in scale (and shift, if applicable) for each image before measuring errors as specified below

- **Scale-invariant point map.** The scale $a^*$ to align prediction with ground truth is computed as:

$$a^* = \underset{a}{\mathrm{argmin}} \sum_{i \in \mathcal{M}} \frac{1}{z_i} \|a\hat{\boldsymbol{p}}_i - \boldsymbol{p}_i\|_1, \qquad (9)$$

- **Affine-invariant point map.** The scale $a^*$ and shift $\boldsymbol{b}^*$ are computed as:

$$(a^*, \boldsymbol{b}^*) = \underset{a, \boldsymbol{b}}{\mathrm{argmin}} \sum_{i \in \mathcal{M}} \frac{1}{z_i} \|a\hat{\boldsymbol{p}}_i + \boldsymbol{b} - \boldsymbol{p}_i\|_1. \qquad (10)$$

- **Scale-invariant depth map**, the scale $a^*$ is computed as

$$a^* = \operatorname*{argmin}_{s} \sum_{i \in \mathcal{M}} \frac{1}{z_i} |a\hat{z}_i - z_i|. \tag{11}$$

- **Affine-invariant depth map.** The scale $a^*$ and shift $b^*$ are computed as

$$(a^*, b^*) = \operatorname*{argmin}_{s} \sum_{i \in \mathcal{M}} \frac{1}{z_i} |a\hat{z}_i + b - z_i|. \tag{12}$$

- **Affine-invariant disparity map.** We follow the established protocol for affine disparity alignment [47], using least-squares to align predictions in disparity space:

$$(a^*, b^*) = \operatorname*{argmin}_{s} \sum_{i \in \mathcal{M}} (a\hat{d}_i + b - d_i)^2, \tag{13}$$

where $\hat{d}_i$ is the predicted disparity and $d_i$ is the ground truth, defined as $d_i = 1/z_i$. To prevent aligned disparities from taking excessively small or negative values, the aligned disparity is truncated by the inverted maximum depth $1/z_{\max}$ before inversion. The final aligned depth $\hat{z}_i^*$ is computed as:

$$\hat{z}_i^* := \frac{1}{\max(a^*\hat{d}_i + b^*, 1/z_{\max})}. \tag{14}$$

**Metric Geometry**

- **Metric depth**. The output is evaluated without alignment and clamping range of values for all methods, unless specific post-processing is hard-coded in its model inference pipeline.

- **Metric point map**. The point map prediction is aligned with the ground truth by the optimal translation:

$$\boldsymbol{b}^* = \operatorname*{argmin}_{\boldsymbol{b}} \sum_{i \in \mathcal{M}} \frac{1}{z_i} \|\hat{\boldsymbol{p}}_i + \boldsymbol{b} - \boldsymbol{p}_i\|_1. \tag{15}$$

## B  Additional Experiments and Results

### B.1  Test-time Resolution Scaling

In ViT-based models, the native input resolution is determined by the number of image tokens derived from fixed-size patches, specifically, $14^2$ for DINOv2 models. As such, resolution scaling can be effectively studied through varying token counts. Our model is trained across a wide range of token counts from 1200 to 3600, corresponding to native input resolutions ranging approximately from $484^2$ to $1188^2$. This training setup enables robust generalization to a broad range of resolutions and flexible usage with details as follows.

**Geometry Accuracy**   MoGe [61] and UniDepth V2 [44] are both trained on diverse input resolutions and aspect ratios, which helps them maintain accuracy under resolution shifts within a moderate range (1200 - 3000). In contrast, models such as Depth Anything [67, 68] and Metric3D V2 [23] are trained with fixed input resolution and exhibit substantial performance degradation when evaluated at resolutions that diverge from their training setting. Our method, trained over a broader resolution spectrum, remains robust under test-time scaling. As shown in Fig. B.3a, it maintains the top accuracy when scaled up for improved detail or down for faster inference—even beyond the training range.

**Boundary Sharpness**   Higher input resolutions and more image tokens generally lead to sharper boundaries in dense prediction tasks, as observed in prior works [68, 48, 29] and also shown in Fig. B.2. In Fig. B.3b, we evaluate several DINOv2-based methods for boundary sharpness at different test-time resolutions. Note that Depth Pro operates at a fixed high resolution due to its specialized multi-scale, patch-based architecture. Our approach consistently delivers the sharpest predictions at each resolution and outperforms Depth Pro using significantly fewer tokens to reach similar levels of detail.

**Latency Trade-off**   As shown in Fig. B.3c, inference latency scales roughly linearly with the number of tokens. Although all compared methods share the same ViT backbone, overall runtime can vary due to differences in decoder complexity and architectural choices. Our model adopts a lightweight design that enables fast inference while maintaining strong accuracy, achieving a favorable trade-off between latency and performance across a wide range of resolutions—within a single unified framework.

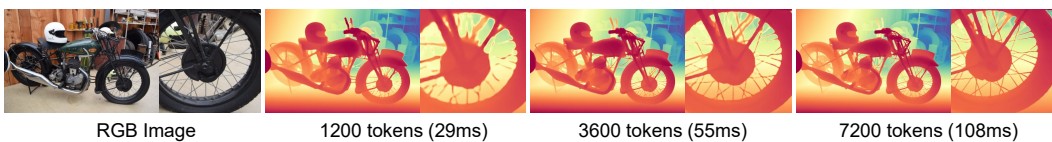

RGB Image        1200 tokens (29ms)        3600 tokens (55ms)        7200 tokens (108ms)

Figure B.2: Trading latency for improved visual sharpness by increasing image tokens.

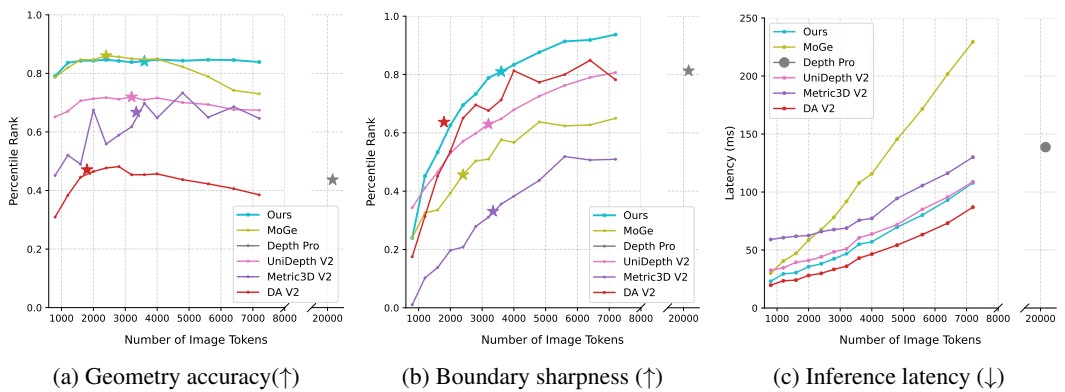

(a) Geometry accuracy(↑)        (b) Boundary sharpness (↑)        (c) Inference latency (↓)

Figure B.3: Performance comparison under test-time resolution scaling. ★ denotes the default configuration for each method. (a) Percentile rank ($\frac{x-\text{worst}}{\text{best}-\text{worst}}$) averaged across all evaluated datasets and two geometry metrics (metric and relative geometry accuracy). (b) Average percentile rank for boundary sharpness. Both are evaluated on a 1/10 subset uniformly sampled from the evaluation benchmarks. (c) Inference latency measured on an NVIDIA A100 GPU with FP16 precision. Our method demonstrates the most favorable balance between latency and performance across different resolutions.

## B.2   Runtime Analysis

As shown in Table B.3, we evaluate the runtime performance of each method under their representative test-time configurations. Specifically, we measure single-frame inference latency and peak GPU memory usage on an NVIDIA A100 GPU. These metrics provide a practical comparison of computational efficiency and resource requirements across different architectures.

## B.3   More Visual Results

More visual results for qualitative comparison are included in Fig. B.4 and Fig. B.5. Representative failure cases are shown in Fig. B.6.

## B.4   Complete Evaluation on Individual Datasets

In the paper, we only listed the average performance across multiple datasets for qualitative comparison and ablation study. Table B.4 and Table B.5 list all the results for each individual datasets.

Table B.3: Runtime statistics measured on a single NVIDIA A100 GPU for single-frame inference.

| Method | #Parameters | #Tokens | Native Resolution | Latency (ms) | | Memory (GB) | |
|---|---|---|---|---|---|---|---|
| | | | | FP16 | FP32 | FP16 | FP32 |
| DA V2 | 335M | 1369 | $518^2$ | 24 | 86 | 0.91 | 1.8 |
| Metric3D V2 | 412M | 3344 | $1064 \times 616$ | 87 | 255 | 1.4 | 2.3 |
| UniDepth V2 | 354M | 1020 | $448^2$ | 33 | 84 | 1.1 | 1.8 |
| | | 3061 | $774^2$ | 50 | 206 | 1.8 | 2.5 |
| Depth Pro | 504M | 20160 | $1536^2$ | 139 | 906 | 3.7 | 8.0 |
| MoGe | 314M | 1200 | $484^2$ | 40 | 93 | 0.74 | 1.4 |
| | | 2500 | $700^2$ | 70 | 192 | 0.88 | 1.6 |
| *Ours* | 326M | 1200 | $484^2$ | 29 | 82 | 0.96 | 1.7 |
| | | 2500 | $700^2$ | 39 | 157 | 1.1 | 2.1 |
| | | 3600 | $840^2$ | 55 | 238 | 1.3 | 2.5 |
| | | 7200 | $1188^2$ | 108 | 565 | 1.9 | 3.8 |

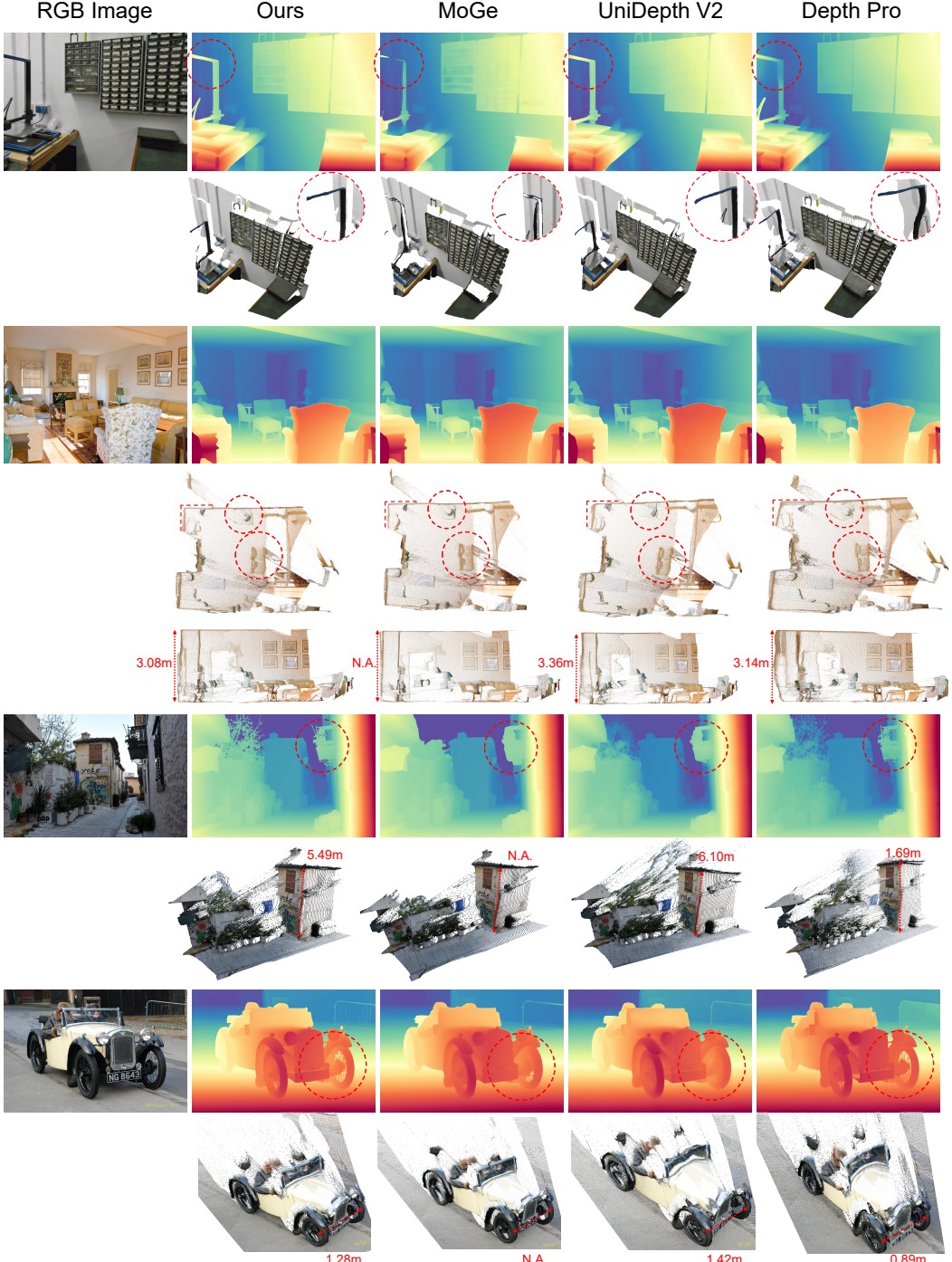

| RGB Image | Ours | MoGe | UniDepth V2 | Depth Pro |

Figure B.4: More visual results on open-domain images (1/2). *Best viewed zoomed in*.

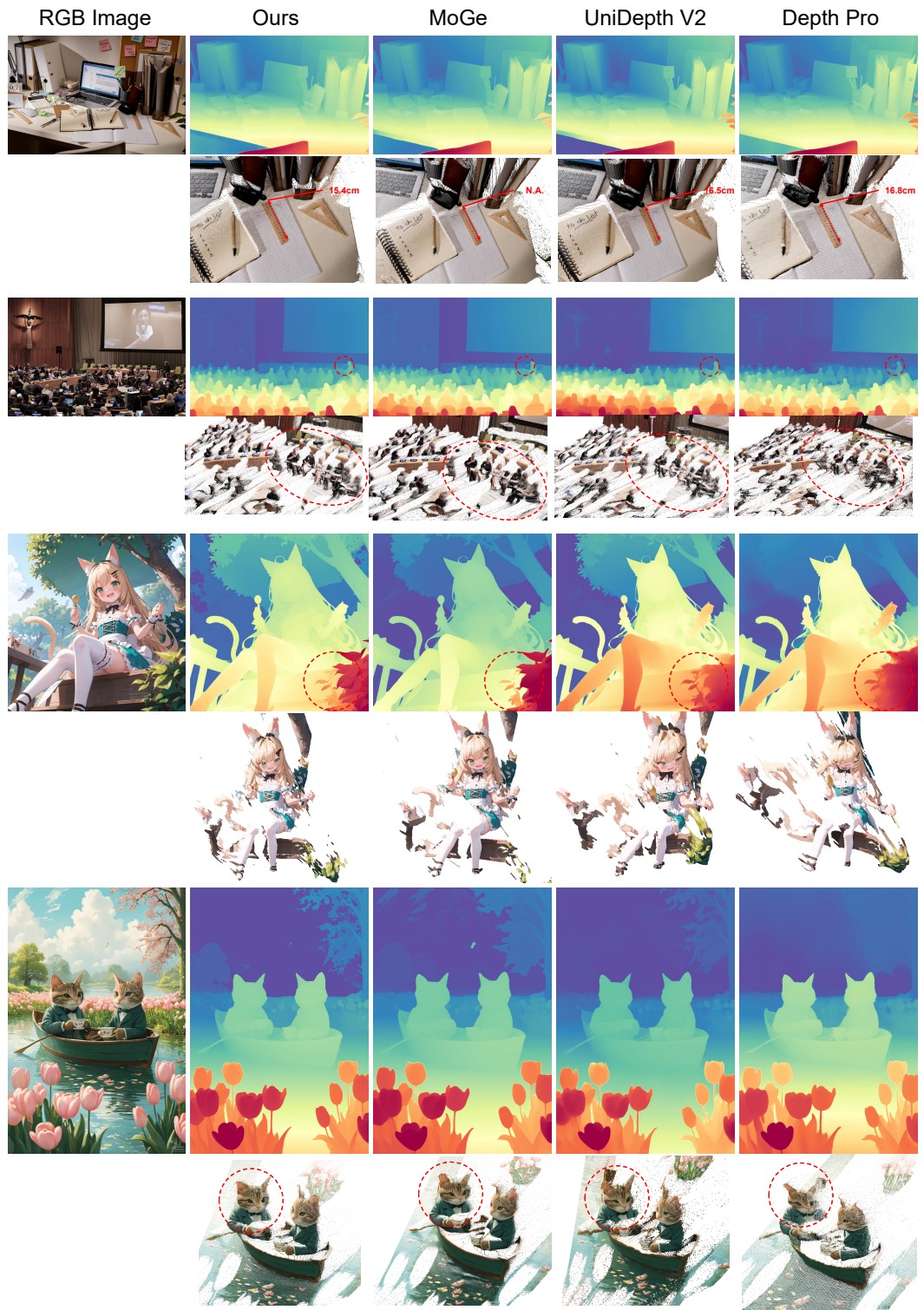

Figure B.5: More visual results on open-domain images (2/2). *Best viewed zoomed in.*

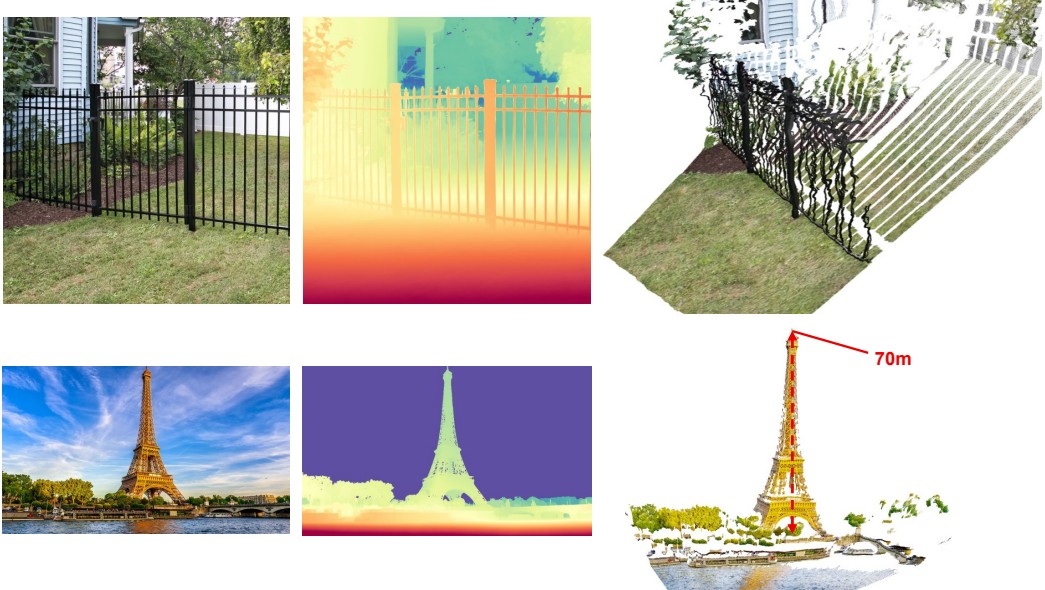

Figure B.6: Failure cases. *Top*: the reconstruction of very thin structures appears distorted, even though the predicted depth maps look sharp. *Bottom*: the model predicts a height of 70 m for the Eiffel Tower, far below its true 330 m. Such scale errors occur when the scene contains atypical content or lacks familiar geometric cues.

Table B.4 evaluation results.

| Method | NYUv2 | | KITTI | | ETH3D | | iBims-1 | | GSO | | Sintel | | DDAD | | DIODE | | Spring | | HAMMER | | *Avg.* | | |
| --- | --- | --- | --- | --- | --- | --- | --- | --- | --- | --- | --- | --- | --- | --- | --- | --- | --- | --- | --- | --- | --- | --- | --- |
| | Rel↓ | $\delta_1$↑ | Rel↓ | $\delta_1$↑ | Rel↓ | $\delta_1$↑ | Rel↓ | $\delta_1$↑ | Rel↓ | $\delta_1$↑ | Rel↓ | $\delta_1$↑ | Rel↓ | $\delta_1$↑ | Rel↓ | $\delta_1$↑ | Rel↓ | $\delta_1$↑ | Rel↓ | $\delta_1$↑ | Rel↓ | $\delta_1$↑ | Rank↓ |
| **Metric point map** | | | | | | | | | | | | | | | | | | | | | | | |
| MASt3R | 7.11 | 95.6 | 26.0 | 45.8 | 27.4 | 43.1 | 10.1 | 89.3 | - | - | - | - | 35.4 | 28.7 | 21.7 | 66.3 | - | - | 56.0 | 18.3 | 26.2 | 55.3 | 4.93 |
| UniDepth V1 | 4.80 | 98.3 | 4.52 | 98.5 | 22.4 | 63.1 | 10.8 | 92.8 | - | - | - | - | 11.4 | 89.5 | 12.8 | 88.9 | - | - | 18.0 | 79.5 | 12.1 | 87.2 | 2.71 |
| UniDepth V2 | 4.83 | 98.0 | 5.88 | 97.5 | 9.46 | 95.0 | 5.23 | 97.9 | - | - | - | - | 13.3 | 90.3 | 17.0 | 80.8 | - | - | 15.0 | 83.9 | 10.1 | 91.9 | 2.43 |
| Depth Pro | 6.13 | 97.3 | 11.1 | 85.3 | 21.2 | 64.9 | 6.89 | 96.9 | - | - | - | - | 22.6 | 61.3 | 13.5 | 81.8 | - | - | 14.5 | 86.0 | 13.7 | 81.9 | 3.29 |
| *Ours* | 4.44 | 98.3 | 7.44 | 94.4 | 7.19 | 97.7 | 5.63 | 97.4 | - | - | - | - | 11.4 | 87.9 | 7.85 | 92.3 | - | - | 13.4 | 87.0 | 8.19 | 93.6 | 1.64 |
| **Metric depth map (wo/ GT intrinsics)** | | | | | | | | | | | | | | | | | | | | | | | |
| ZoeDepth | 11.0 | 91.9 | 17.0 | 85.4 | 57.1 | 33.7 | 17.4 | 67.2 | - | - | - | - | 38.9 | 38.6 | 39.3 | 29.3 | - | - | 94.3 | 3.23 | 39.3 | 49.9 | 5.90 |
| MASt3R | 10.8 | 89.7 | 56.7 | 9.84 | 47.2 | 20.1 | 18.7 | 61.5 | - | - | - | - | 62.4 | 5.51 | 54.9 | 19.0 | - | - | 97.2 | 6.74 | 49.7 | 30.3 | 6.71 |
| DA V1 | 10.5 | 94.9 | 11.6 | 94.5 | 40.2 | 24.0 | 12.9 | 81.8 | - | - | - | - | 34.5 | 44.7 | 58.0 | 16.2 | - | - | 54.8 | 27.3 | 31.8 | 54.8 | 5.50 |
| DA V2 | 16.4 | 80.9 | 10.6 | 88.6 | 36.1 | 36.3 | 11.1 | 91.7 | - | - | - | - | 41.7 | 37.5 | 41.2 | 22.1 | - | - | 52.1 | 38.9 | 29.9 | 56.6 | 4.43 |
| UniDepth V1 | 7.59 | 97.6 | 4.69 | 98.4 | 56.9 | 14.9 | 23.8 | 57.6 | - | - | - | - | 13.8 | 85.1 | 17.1 | 71.9 | - | - | 38.2 | 46.7 | 23.2 | 67.5 | 3.32 |
| UniDepth V2 | 10.6 | 92.8 | 8.58 | 95.4 | 20.7 | 69.5 | 9.52 | 93.2 | - | - | - | - | 18.4 | 77.6 | 43.0 | 51.8 | - | - | 38.2 | 46.8 | 21.3 | 75.3 | 2.54 |
| Depth Pro | 10.7 | 91.9 | 23.5 | 38.3 | 38.5 | 32.8 | 15.9 | 81.5 | - | - | - | - | 33.4 | 35.3 | 31.9 | 37.7 | - | - | 39.1 | 63.0 | 27.6 | 54.4 | 4.36 |
| *Ours* | 7.33 | 96.1 | 18.1 | 62.9 | 10.4 | 90.8 | 13.6 | 83.0 | - | - | - | - | 15.8 | 73.0 | 17.5 | 66.4 | - | - | 26.9 | 65.6 | 15.7 | 76.8 | 2.21 |
| **Metric depth map (w/ GT intrinsics)** | | | | | | | | | | | | | | | | | | | | | | | |
| Metric3D V2 | 7.16 | 96.5 | 5.25 | 98.0 | 11.8 | 88.8 | 9.96 | 94.1 | - | - | - | - | 9.21 | 93.7 | 49.1 | 1.98 | - | - | 35.7 | 44.3 | 18.3 | 73.9 | 2.75 |
| UniDepth V1 | 5.98 | 97.9 | 4.43 | 98.5 | 44.5 | 26.7 | 22.6 | 60.5 | - | - | - | - | 13.0 | 87.2 | 21.0 | 63.5 | - | - | 38.6 | 45.9 | 21.4 | 68.6 | 2.50 |
| UniDepth V2 | 7.81 | 96.0 | 5.98 | 97.7 | 15.0 | 85.2 | 7.71 | 95.5 | - | - | - | - | 14.1 | 89.3 | 41.0 | 67.1 | - | - | 37.7 | 47.1 | 18.5 | 82.6 | 2.57 |
| *Ours* | 6.46 | 96.9 | 8.64 | 93.7 | 10.5 | 92.2 | 9.92 | 92.4 | - | - | - | - | 13.1 | 85.6 | 16.2 | 77.1 | - | - | 30.4 | 74.2 | 13.6 | 87.4 | 2.00 |
| **Scale-invariant point map** | | | | | | | | | | | | | | | | | | | | | | | |
| MASt3R | 6.26 | 96.0 | 10.0 | 93.8 | 6.28 | 95.5 | 7.55 | 95.1 | 5.03 | 99.0 | 31.5 | 50.2 | 15.9 | 77.6 | 12.8 | 85.0 | 39.3 | 33.7 | 10.7 | 95.0 | 14.5 | 82.1 | 5.45 |
| UniDepth V1 | 5.33 | 98.4 | 5.96 | 98.5 | 18.5 | 77.6 | 5.29 | 97.4 | 6.58 | 99.6 | 33.0 | 48.9 | 11.4 | 90.2 | 12.3 | 91.0 | 33.1 | 49.8 | 4.83 | 98.5 | 13.6 | 85.0 | 3.83 |
| UniDepth V2 | 5.59 | 98.3 | 5.41 | 98.0 | 6.58 | 97.2 | 5.56 | 98.1 | 4.53 | 99.7 | 27.2 | 56.3 | 13.4 | 91.2 | 12.0 | 93.4 | 31.9 | 46.0 | 4.20 | 99.2 | 11.6 | 87.7 | 2.98 |
| Depth Pro | 5.04 | 97.7 | 10.6 | 95.1 | 11.2 | 92.0 | 5.84 | 97.1 | 4.94 | 99.8 | 26.9 | 43.9 | 15.8 | 81.0 | 8.52 | 91.6 | 28.1 | 60.5 | 6.82 | 98.7 | 12.4 | 87.7 | 3.83 |
| MoGe | 4.86 | 98.4 | 5.47 | 97.4 | 4.58 | 98.9 | 4.63 | 97.1 | 2.58 | 100 | 22.3 | 69.5 | 12.3 | 90.3 | 6.58 | 94.5 | 4.84 | 96.4 | 6.45 | 98.1 | 7.46 | 94.1 | 2.14 |
| *Ours* | 3.94 | 98.3 | 8.27 | 97.5 | 5.45 | 98.6 | 5.34 | 98.3 | 2.55 | 100 | 23.1 | 66.8 | 11.0 | 90.7 | 8.42 | 93.7 | 31.1 | 42.4 | 8.77 | 98.4 | 10.8 | 88.5 | 2.40 |
| **Affine-invariant point map** | | | | | | | | | | | | | | | | | | | | | | | |
| MASt3R | 5.30 | 96.3 | 8.32 | 92.3 | 5.48 | 96.6 | 5.72 | 95.0 | 3.50 | 99.2 | 26.3 | 62.8 | 14.7 | 79.6 | 8.10 | 90.1 | 33.3 | 51.1 | 5.34 | 96.6 | 11.6 | 86.0 | 5.45 |
| UniDepth V1 | 3.93 | 98.4 | 4.29 | 98.6 | 12.2 | 89.6 | 4.65 | 98.0 | 2.99 | 99.8 | 28.5 | 58.4 | 10.3 | 90.5 | 8.56 | 90.9 | 29.6 | 58.5 | 4.15 | 98.7 | 10.9 | 88.1 | 3.95 |
| UniDepth V2 | 3.66 | 98.4 | 4.75 | 98.0 | 4.35 | 98.4 | 4.05 | 98.1 | 2.91 | 99.9 | 17.9 | 76.5 | 12.0 | 90.8 | 7.45 | 92.4 | 25.1 | 66.9 | 3.45 | 99.4 | 8.56 | 91.9 | 2.55 |
| Depth Pro | 4.36 | 97.9 | 9.15 | 90.7 | 7.73 | 94.0 | 4.34 | 97.4 | 3.16 | 99.7 | 19.6 | 74.5 | 14.4 | 81.2 | 6.28 | 93.7 | 25.0 | 66.0 | 5.31 | 98.8 | 9.93 | 89.4 | 4.30 |
| MoGe | 3.68 | 98.3 | 4.86 | 97.2 | 3.57 | 99.0 | 3.61 | 97.3 | 1.14 | 100 | 16.8 | 77.8 | 10.5 | 91.4 | 4.37 | 96.4 | 4.45 | 96.4 | 3.88 | 98.1 | 5.69 | 95.2 | 2.14 |
| *Ours* | 3.33 | 98.4 | 6.47 | 96.4 | 3.89 | 98.7 | 3.65 | 98.5 | 1.16 | 100 | 17.4 | 77.0 | 11.0 | 90.3 | 5.13 | 94.9 | 24.5 | 63.7 | 4.19 | 99.1 | 7.98 | 91.7 | 2.23 |
| **Local point map** | | | | | | | | | | | | | | | | | | | | | | | |
| MASt3R | - | - | - | - | 5.54 | 95.3 | 6.19 | 95.0 | - | - | 11.4 | 87.9 | 8.58 | 91.8 | 8.75 | 90.9 | - | - | - | - | 8.09 | 92.2 | 5.40 |
| UniDepth V1 | - | - | - | - | 8.61 | 92.6 | 5.92 | 96.0 | - | - | 13.4 | 84.3 | 8.18 | 92.0 | 9.95 | 90.0 | - | - | - | - | 9.21 | 91.0 | 5.55 |
| UniDepth V2 | - | - | - | - | 3.99 | 97.4 | 4.02 | 97.3 | - | - | 9.35 | 92.2 | 8.18 | 92.4 | 6.15 | 95.3 | - | - | - | - | 6.34 | 94.9 | 3.10 |
| Depth Pro | - | - | - | - | 4.76 | 96.9 | 4.11 | 97.5 | - | - | 10.8 | 89.5 | 8.08 | 92.4 | 6.80 | 94.1 | - | - | - | - | 6.91 | 94.1 | 3.55 |
| MoGe | - | - | - | - | 3.21 | 98.1 | 4.16 | 96.8 | - | - | 8.63 | 92.7 | 6.74 | 94.3 | 4.78 | 96.3 | - | - | - | - | 5.50 | 95.6 | 2.05 |
| *Ours* | - | - | - | - | 3.27 | 98.2 | 3.61 | 97.7 | - | - | 8.13 | 93.2 | 6.57 | 94.5 | 5.09 | 96.1 | - | - | - | - | 5.33 | 95.9 | 1.35 |
| **Scale-invariant depth map** | | | | | | | | | | | | | | | | | | | | | | | |
| ZoeDepth | 5.62 | 96.3 | 7.27 | 91.9 | 10.4 | 87.3 | 7.45 | 93.2 | 3.23 | 99.9 | 27.4 | 61.8 | 17.0 | 72.8 | 11.3 | 85.2 | 30.3 | 55.9 | 7.42 | 94.7 | 12.7 | 83.9 | 8.75 |
| MASt3R | 5.37 | 96.0 | 6.24 | 94.5 | 5.68 | 95.5 | 5.58 | 95.2 | 3.72 | 99.1 | 26.3 | 63.7 | 13.5 | 81.5 | 8.37 | 89.4 | 32.2 | 53.5 | 5.50 | 96.5 | 11.2 | 86.5 | 7.65 |
| DA V1 | 4.77 | 97.5 | 5.61 | 95.6 | 9.41 | 88.9 | 5.53 | 95.8 | 5.49 | 99.3 | 28.3 | 56.7 | 13.2 | 81.5 | 10.3 | 87.5 | 27.3 | 59.1 | 6.88 | 96.4 | 11.7 | 85.8 | 8.22 |
| DA V2 | 5.03 | 97.3 | 7.23 | 93.7 | 6.12 | 95.5 | 4.32 | 97.9 | 4.38 | 99.3 | 23.0 | 65.2 | 14.7 | 78.0 | 7.95 | 90.0 | 28.0 | 61.1 | 5.92 | 97.7 | 10.7 | 87.6 | 6.80 |
| Metric3D V2 | 4.69 | 97.4 | 4.00 | 98.5 | 3.84 | 98.5 | 4.23 | 97.7 | 2.46 | 99.9 | 20.7 | 69.8 | 7.41 | 94.6 | 3.29 | 98.4 | 24.4 | 64.4 | 4.19 | 99.1 | 7.92 | 91.8 | 3.39 |
| UniDepth V1 | 3.86 | 98.4 | 3.73 | 98.6 | 5.67 | 97.0 | 4.79 | 97.4 | 4.18 | 99.7 | 28.3 | 58.8 | 10.1 | 90.5 | 6.83 | 92.8 | 29.2 | 59.3 | 4.19 | 98.4 | 10.1 | 89.1 | 5.12 |
| UniDepth V2 | 3.65 | 98.4 | 4.24 | 98.0 | 3.23 | 98.9 | 3.45 | 98.1 | 3.16 | 99.7 | 23.1 | 65.3 | 11.0 | 91.5 | 5.92 | 94.1 | 24.9 | 65.1 | 3.48 | 99.1 | 8.61 | 90.8 | 3.10 |
| Depth Pro | 4.42 | 97.6 | 5.47 | 96.2 | 7.54 | 94.1 | 4.13 | 97.4 | 2.18 | 99.9 | 23.9 | 68.7 | 14.0 | 82.0 | 7.05 | 92.0 | 25.1 | 63.8 | 4.36 | 98.9 | 9.81 | 89.1 | 5.33 |
| MoGe | 3.44 | 98.4 | 4.25 | 97.8 | 3.36 | 98.9 | 3.46 | 97.0 | 1.47 | 100 | 19.3 | 73.4 | 9.17 | 90.5 | 4.89 | 94.7 | 4.63 | 96.4 | 3.77 | 98.1 | 5.77 | 94.5 | 2.72 |
| *Ours* | 3.44 | 98.2 | 4.11 | 98.0 | 3.55 | 98.7 | 3.16 | 98.2 | 1.49 | 100 | 19.6 | 71.6 | 8.91 | 91.2 | 5.30 | 94.6 | 20.0 | 72.4 | 3.96 | 99.2 | 7.35 | 92.2 | 2.12 |
| **Affine-invariant depth** | | | | | | | | | | | | | | | | | | | | | | | |
| ZoeDepth | 4.76 | 97.3 | 5.59 | 95.1 | 7.27 | 94.2 | 5.85 | 95.7 | 2.54 | 99.9 | 21.8 | 69.2 | 14.2 | 80.1 | 7.80 | 90.9 | 24.3 | 66.6 | 6.65 | 95.7 | 10.1 | 88.5 | 9.09 |
| MASt3R | 4.67 | 96.7 | 5.79 | 95.1 | 4.64 | 97.0 | 4.62 | 95.6 | 2.85 | 99.4 | 21.3 | 70.3 | 12.5 | 83.4 | 5.79 | 94.1 | 27.4 | 62.8 | 4.21 | 96.8 | 9.38 | 89.1 | 7.97 |
| DA V1 | 3.82 | 98.3 | 5.04 | 96.4 | 6.23 | 95.2 | 4.23 | 97.3 | 1.98 | 100 | 20.1 | 71.8 | 11.3 | 86.1 | 6.75 | 92.6 | 22.4 | 68.9 | 5.77 | 97.3 | 8.76 | 90.4 | 6.91 |
| DA V2 | 4.16 | 97.9 | 6.77 | 94.3 | 4.63 | 97.2 | 3.44 | 98.3 | 1.44 | 100 | 17.1 | 76.6 | 13.4 | 81.8 | 5.41 | 94.6 | 23.7 | 68.7 | 4.73 | 98.9 | 8.48 | 90.8 | 6.15 |
| Metric3D V2 | 3.94 | 97.6 | 3.50 | 98.4 | 3.24 | 99.0 | 3.28 | 98.3 | 2.10 | 99.4 | 26.6 | 71.7 | 7.15 | 94.8 | 2.75 | 98.7 | 21.0 | 72.5 | 3.02 | 99.0 | 7.66 | 92.9 | 4.53 |
| UniDepth V1 | 3.40 | 98.6 | 3.55 | 98.7 | 4.92 | 97.5 | 3.76 | 98.2 | 2.48 | 99.9 | 24.9 | 64.1 | 9.46 | 90.8 | 4.90 | 96.2 | 25.3 | 67.3 | 3.55 | 98.9 | 8.61 | 91.0 | 5.67 |
| UniDepth V2 | 2.96 | 98.6 | 3.85 | 98.1 | 2.95 | 98.5 | 2.64 | 98.4 | 1.37 | 100 | 13.3 | 83.2 | 10.5 | 90.9 | 4.05 | 96.5 | 20.1 | 75.4 | 2.48 | 99.6 | 6.42 | 93.9 | 2.80 |
| Depth Pro | 3.67 | 98.2 | 5.12 | 96.8 | 4.97 | 96.4 | 3.23 | 98.3 | 1.46 | 100 | 15.8 | 80.1 | 12.6 | 84.1 | 4.66 | 95.6 | 21.7 | 70.5 | 3.30 | 99.6 | 7.65 | 92.0 | 5.05 |
| MoGe | 2.92 | 98.6 | 3.94 | 98.0 | 2.69 | 99.2 | 2.74 | 97.9 | 0.94 | 100 | 13.0 | 83.2 | 8.40 | 92.1 | 3.16 | 97.5 | 4.34 | 96.4 | 3.00 | 98.3 | 4.51 | 96.1 | 2.94 |
| *Ours* | 2.89 | 98.6 | 3.75 | 98.1 | 2.80 | 99.1 | 2.36 | 98.8 | 0.94 | 100 | 13.3 | 82.5 | 8.26 | 92.5 | 3.14 | 97.4 | 15.9 | 81.2 | 2.85 | 99.3 | 5.62 | 94.8 | 2.02 |
| **Affine-invariant disparity** | | | | | | | | | | | | | | | | | | | | | | | |
| ZoeDepth | 5.21 | 97.7 | 5.84 | 95.6 | 8.07 | 94.0 | 6.19 | 96.1 | 2.60 | 99.9 | 26.9 | 66.3 | 14.1 | 81.7 | 8.17 | 92.0 | 27.2 | 63.0 | 6.84 | 96.4 | 11.1 | 88.3 | 8.78 |
| DA V1 | 4.20 | 98.4 | 5.40 | 97.0 | 4.68 | 98.2 | 4.18 | 97.6 | 1.54 | 100 | 20.2 | 77.6 | 12.7 | 86.9 | 5.69 | 95.7 | 22.2 | 72.5 | 5.56 | 98.0 | 8.63 | 92.2 | 5.62 |
| DA V2 | 4.14 | 98.3 | 5.61 | 96.7 | 4.71 | 97.9 | 3.47 | 98.5 | 1.24 | 100 | 21.4 | 72.8 | 13.1 | 86.4 | 5.29 | 96.1 | 24.3 | 70.6 | 4.97 | 99.1 | 8.82 | 91.6 | 5.42 |
| Metric3D V2 | 13.4 | 81.5 | 3.76 | 98.2 | 4.30 | 97.7 | 8.55 | 92.3 | 1.80 | 100 | 21.8 | 72.4 | 7.35 | 94.1 | 7.70 | 90.2 | 23.3 | 68.1 | 3.17 | 99.2 | 9.51 | 89.4 | 6.17 |
| MASt3R | 5.07 | 96.8 | 5.93 | 95.5 | 5.25 | 96.4 | 5.39 | 95.7 | 2.98 | 99.7 | 30.2 | 65.1 | 13.0 | 83.6 | 6.41 | 94.3 | 37.3 | 53.2 | 4.41 | 97.2 | 11.6 | 87.8 | 8.60 |
| UniDepth V1 | 3.78 | 98.7 | 3.64 | 98.7 | 5.34 | 97.2 | 4.06 | 98.1 | 2.56 | 99.9 | 28.6 | 60.7 | 9.94 | 89.1 | 5.95 | 95.5 | 30.0 | 61.6 | 3.64 | 99.1 | 9.75 | 89.9 | 5.92 |
| UniDepth V2 | 3.38 | 98.7 | 3.99 | 98.0 | 2.97 | 99.0 | 3.15 | 98.3 | 1.30 | 100 | 17.2 | 79.9 | 10.2 | 90.2 | 4.43 | 96.4 | 24.4 | 69.6 | 2.51 | 99.6 | 7.35 | 93.0 | 2.75 |
| Depth Pro | 4.21 | 98.1 | 5.10 | 97.0 | 4.94 | 96.7 | 3.74 | 98.2 | 1.49 | 100 | 18.4 | 79.5 | 11.7 | 87.1 | 4.84 | 96.2 | 27.5 | 64.5 | 3.31 | 99.6 | 8.42 | 91.7 | 5.08 |
| MoGe | 3.38 | 98.6 | 4.05 | 98.1 | 3.11 | 98.9 | 3.23 | 98.0 | 0.96 | 100 | 18.4 | 79.5 | 8.99 | 91.5 | 3.98 | 97.2 | 6.43 | 93.7 | 3.30 | 98.5 | 5.58 | 95.4 | 3.17 |
| *Ours* | 3.35 | 98.6 | 3.92 | 98.1 | 3.21 | 98.9 | 2.85 | 98.7 | 0.96 | 100 | 18.0 | 78.7 | 8.69 | 92.1 | 4.03 | 97.2 | 18.7 | 76.6 | 2.90 | 99.5 | 6.66 | 93.8 | 2.17 |

Table B.4: Evaluation results of baselines and our method on each dataset.

| Ablation | | NYUv2 | | KITTI | | ETH3D | | iBims-1 | | GSO | | Sintel | | DDAD | | DIODE | | Spring | | HAMMER | | *Avg.* | |
|---|---|---|---|---|---|---|---|---|---|---|---|---|---|---|---|---|---|---|---|---|---|---|---|
| Data | Scale Prediction | Rel↓ | δ₁↑ | Rel↓ | δ₁↑ | Rel↓ | δ₁↑ | Rel↓ | δ₁↑ | Rel↓ | δ₁↑ | Rel↓ | δ₁↑ | Re↓ | δ₁↑ | Rel↓ | δ₁↑ | Rel↓ | δ₁↑ | Rel↓ | δ₁↑ | Rel↓ | δ₁↑ |
| **Metric point map** | | | | | | | | | | | | | | | | | | | | | | | |
| Improved real | Entangled (SI-Log) | 6.00 | 97.3 | 8.33 | 93.4 | 11.6 | 89.4 | 7.78 | 94.6 | - | - | - | - | 14.4 | 83.1 | 10.6 | 88.4 | - | - | 11.4 | 88.7 | 10.0 | 90.7 |
| Improved real | Entangled (Shift inv.) | 5.26 | 97.6 | 8.81 | 92.6 | 10.4 | 91.7 | 6.14 | 96.6 | - | - | - | - | 13.0 | 84.8 | 8.97 | 91.0 | - | - | 10.4 | 90.3 | 9.00 | 92.1 |
| Improved real | Decoupled (Conv) | 5.37 | 97.8 | 9.56 | 91.8 | 9.46 | 94.1 | 6.49 | 95.6 | - | - | - | - | 13.3 | 83.5 | 8.93 | 91.9 | - | - | 14.2 | 85.1 | 9.62 | 91.4 |
| Synthetic only | Decoupled (MLP) | 8.58 | 94.7 | 9.48 | 91.9 | 14.9 | 83.4 | 8.20 | 94.2 | - | - | - | - | 16.5 | 80.4 | 11.0 | 88.8 | - | - | 18.4 | 78.2 | 12.4 | 87.4 |
| Raw real | Decoupled (MLP) | 5.36 | 97.8 | 7.70 | 94.5 | 8.58 | 94.6 | 6.60 | 95.7 | - | - | - | - | 12.2 | 85.5 | 9.01 | 91.5 | - | - | 13.7 | 85.4 | 9.02 | 92.1 |
| Improved real | Decoupled (MLP) | 5.47 | 97.6 | 8.98 | 92.6 | 8.75 | 94.3 | 6.24 | 96.1 | - | - | - | - | 12.8 | 84.6 | 9.26 | 90.9 | - | - | 12.9 | 87.4 | 9.20 | 91.9 |
| **Metric depth map (wo/ GT intrinsics)** | | | | | | | | | | | | | | | | | | | | | | | |
| Improved real | Entangled (SI-Log) | 9.65 | 91.4 | 14.5 | 77.3 | 16.4 | 73.7 | 20.1 | 56.2 | - | - | - | - | 19.1 | 67.7 | 22.3 | 54.3 | - | - | 23.0 | 59.8 | 17.9 | 68.6 |
| Improved real | Entangled (Shift inv.) | 9.04 | 93.1 | 19.1 | 56.8 | 15.5 | 76.8 | 15.1 | 72.1 | - | - | - | - | 18.0 | 67.9 | 19.9 | 59.8 | - | - | 22.0 | 55.3 | 16.9 | 68.8 |
| Improved real | Decoupled (Conv) | 9.22 | 92.7 | 20.3 | 51.8 | 13.8 | 79.8 | 15.8 | 71.0 | - | - | - | - | 18.1 | 66.6 | 19.0 | 61.8 | - | - | 27.5 | 54.9 | 17.7 | 68.4 |
| Synthetic only | Decoupled (MLP) | 18.1 | 73.7 | 15.8 | 71.0 | 24.7 | 53.2 | 15.8 | 76.6 | - | - | - | - | 21.9 | 62.4 | 22.7 | 56.5 | - | - | 32.8 | 62.0 | 21.7 | 65.1 |
| Raw real | Decoupled (MLP) | 9.22 | 92.9 | 13.8 | 80.5 | 13.8 | 82.1 | 16.7 | 72.4 | - | - | - | - | 16.5 | 72.3 | 19.7 | 61.5 | - | - | 20.8 | 67.9 | 15.8 | 75.7 |
| Improved real | Decoupled (MLP) | 9.48 | 92.2 | 18.7 | 59.2 | 13.5 | 82.6 | 13.6 | 79.3 | - | - | - | - | 17.0 | 69.2 | 20.0 | 59.7 | - | - | 23.4 | 67.4 | 16.5 | 72.8 |
| **Scale-invariant point map** | | | | | | | | | | | | | | | | | | | | | | | |
| Improved real | Entangled (SI-Log) | 6.03 | 97.4 | 9.68 | 95.3 | 8.13 | 95.3 | 8.63 | 96.8 | 4.01 | 100 | 26.6 | 59.4 | 13.8 | 84.8 | 10.3 | 89.8 | 31.2 | 48.0 | 10.4 | 95.6 | 12.9 | 86.2 |
| Improved real | Entangled (Shift inv.) | 5.00 | 97.8 | 10.7 | 95.8 | 7.02 | 96.7 | 7.42 | 97.4 | 3.42 | 100 | 26.0 | 58.2 | 12.7 | 86.6 | 8.97 | 92.2 | 28.9 | 49.8 | 10.5 | 97.5 | 12.1 | 87.2 |
| Improved real | Decoupled (Conv) | 4.84 | 97.8 | 12.1 | 94.8 | 6.55 | 96.9 | 7.15 | 96.9 | 3.19 | 100 | 26.3 | 55.8 | 12.9 | 85.5 | 9.11 | 91.8 | 29.9 | 46.9 | 10.4 | 97.0 | 12.2 | 86.3 |
| Synthetic only | Decoupled (MLP) | 6.66 | 96.9 | 11.3 | 93.0 | 6.85 | 95.8 | 5.99 | 96.7 | 3.14 | 100 | 25.4 | 61.3 | 15.0 | 81.1 | 10.5 | 89.4 | 30.9 | 46.8 | 7.39 | 97.5 | 12.3 | 85.8 |
| Raw real | Decoupled (MLP) | 4.88 | 98.0 | 9.15 | 96.0 | 6.08 | 97.1 | 7.31 | 96.8 | 3.06 | 100 | 24.8 | 60.8 | 11.8 | 87.7 | 8.34 | 92.3 | 28.1 | 53.2 | 10.9 | 96.2 | 11.4 | 87.8 |
| Improved real | Decoupled (MLP) | 5.00 | 97.8 | 11.2 | 95.0 | 6.21 | 97.4 | 6.52 | 97.3 | 2.97 | 100 | 25.6 | 60.3 | 12.6 | 87.0 | 8.76 | 92.3 | 28.3 | 51.0 | 9.10 | 98.3 | 11.6 | 87.6 |
| **Affine-invariant point map** | | | | | | | | | | | | | | | | | | | | | | | |
| Improved real | Entangled (SI-Log) | 5.00 | 97.7 | 8.22 | 93.0 | 6.72 | 96.1 | 5.71 | 96.8 | 2.57 | 100 | 21.1 | 71.0 | 12.9 | 84.5 | 7.36 | 91.9 | 26.7 | 59.7 | 6.37 | 97.1 | 10.3 | 88.8 |
| Improved real | Entangled (Shift inv.) | 4.17 | 97.9 | 8.58 | 93.0 | 5.42 | 97.0 | 4.74 | 96.8 | 1.78 | 100 | 19.7 | 72.7 | 11.7 | 86.6 | 6.07 | 93.5 | 23.3 | 66.3 | 5.02 | 98.6 | 9.05 | 90.2 |
| Improved real | Decoupled (Conv) | 4.08 | 98.0 | 9.65 | 90.8 | 5.12 | 97.0 | 4.67 | 97.0 | 1.66 | 100 | 19.6 | 72.2 | 12.0 | 85.4 | 6.11 | 93.4 | 23.4 | 67.9 | 5.24 | 98.0 | 9.15 | 90.0 |
| Synthetic only | Decoupled (MLP) | 5.48 | 97.0 | 9.11 | 90.2 | 5.93 | 96.2 | 4.94 | 96.6 | 1.56 | 100 | 20.0 | 73.1 | 13.7 | 81.9 | 7.01 | 91.6 | 25.5 | 63.7 | 4.44 | 98.7 | 9.77 | 88.9 |
| Raw real | Decoupled (MLP) | 4.06 | 98.2 | 7.37 | 94.5 | 4.89 | 97.5 | 4.74 | 96.8 | 1.61 | 100 | 19.0 | 73.6 | 10.9 | 87.9 | 5.89 | 93.7 | 23.4 | 67.0 | 5.09 | 98.2 | 8.70 | 90.7 |
| Improved real | Decoupled (MLP) | 4.14 | 98.0 | 8.95 | 92.0 | 4.94 | 97.5 | 4.50 | 97.2 | 1.62 | 100 | 19.6 | 73.6 | 11.7 | 86.6 | 6.06 | 93.3 | 22.8 | 68.7 | 4.40 | 98.9 | 8.87 | 90.6 |
| **Local point map** | | | | | | | | | | | | | | | | | | | | | | | |
| Improved real | Entangled (SI-Log) | - | - | - | - | 6.30 | 95.6 | 5.96 | 96.6 | - | - | 12.0 | 87.5 | 8.14 | 92.5 | 8.63 | 92.5 | - | - | - | - | 8.21 | 92.9 |
| Improved real | Entangled (Shift inv.) | - | - | - | - | 4.61 | 97.2 | 4.56 | 97.2 | - | - | 10.3 | 90.2 | 7.38 | 93.5 | 6.61 | 94.7 | - | - | - | - | 6.69 | 94.6 |
| Improved real | Decoupled (Conv) | - | - | - | - | 4.25 | 97.5 | 4.34 | 97.3 | - | - | 9.72 | 91.0 | 7.24 | 93.6 | 6.17 | 95.1 | - | - | - | - | 6.34 | 94.9 |
| Synthetic only | Decoupled (MLP) | - | - | - | - | 4.37 | 97.4 | 4.45 | 97.2 | - | - | 9.33 | 91.7 | 7.51 | 93.3 | 6.44 | 94.9 | - | - | - | - | 6.42 | 94.9 |
| Raw real | Decoupled (MLP) | - | - | - | - | 4.28 | 97.4 | 4.55 | 97.1 | - | - | 9.64 | 91.2 | 7.11 | 93.7 | 6.28 | 94.9 | - | - | - | - | 6.37 | 94.9 |
| Improved real | Decoupled (MLP) | - | - | - | - | 4.20 | 97.5 | 4.31 | 97.3 | - | - | 9.34 | 91.9 | 7.21 | 93.7 | 6.21 | 95.0 | - | - | - | - | 6.25 | 95.1 |
| **Scale-invariant depth map** | | | | | | | | | | | | | | | | | | | | | | | |
| Improved real | Entangled (SI-Log) | 4.99 | 97.4 | 5.18 | 96.7 | 6.48 | 95.2 | 5.26 | 97.3 | 2.64 | 100 | 23.2 | 65.7 | 11.6 | 86.2 | 7.76 | 91.3 | 24.8 | 63.4 | 6.40 | 96.7 | 9.83 | 89.0 |
| Improved real | Entangled (Shift inv.) | 4.17 | 97.8 | 4.57 | 97.5 | 5.03 | 96.7 | 4.42 | 97.6 | 2.09 | 100 | 22.4 | 66.7 | 10.3 | 88.0 | 6.16 | 93.5 | 20.8 | 69.2 | 4.69 | 98.6 | 8.46 | 90.6 |
| Improved real | Decoupled (Conv) | 4.08 | 97.9 | 4.61 | 97.3 | 4.80 | 96.9 | 4.32 | 97.1 | 1.92 | 100 | 22.5 | 64.9 | 10.3 | 87.7 | 6.26 | 93.2 | 21.0 | 68.8 | 4.81 | 98.2 | 8.46 | 90.2 |
| Synthetic only | Decoupled (MLP) | 5.05 | 96.9 | 5.47 | 96.3 | 5.64 | 95.9 | 4.76 | 96.9 | 1.90 | 100 | 21.7 | 68.0 | 12.0 | 85.1 | 7.16 | 91.2 | 22.2 | 67.4 | 4.66 | 97.9 | 9.05 | 89.6 |
| Raw real | Decoupled (MLP) | 4.09 | 98.0 | 4.60 | 97.2 | 4.82 | 97.1 | 4.49 | 97.0 | 1.92 | 100 | 21.8 | 66.5 | 9.79 | 88.4 | 6.09 | 93.2 | 21.8 | 68.6 | 4.67 | 98.1 | 8.41 | 90.4 |
| Improved real | Decoupled (MLP) | 4.16 | 97.8 | 4.59 | 97.2 | 4.62 | 97.4 | 4.20 | 97.4 | 1.89 | 100 | 21.9 | 67.8 | 10.1 | 88.4 | 6.08 | 93.3 | 20.4 | 71.6 | 4.34 | 98.9 | 8.23 | 91.0 |
| **Affine-invariant depth** | | | | | | | | | | | | | | | | | | | | | | | |
| Improved real | Entangled (SI-Log) | 4.32 | 98.0 | 4.91 | 97.0 | 5.21 | 96.9 | 4.15 | 97.8 | 2.11 | 100 | 17.4 | 76.3 | 10.8 | 87.9 | 5.17 | 95.7 | 20.3 | 72.6 | 5.26 | 97.6 | 7.96 | 92.0 |
| Improved real | Entangled (Shift inv.) | 3.57 | 98.2 | 4.26 | 97.6 | 3.98 | 97.9 | 3.35 | 98.2 | 1.44 | 100 | 16.0 | 78.4 | 9.58 | 89.7 | 4.01 | 96.8 | 17.7 | 76.7 | 3.58 | 98.7 | 6.75 | 93.2 |
| Improved real | Decoupled (Conv) | 3.48 | 98.3 | 4.31 | 97.4 | 3.77 | 98.1 | 3.23 | 98.0 | 1.35 | 100 | 15.2 | 79.2 | 9.48 | 89.5 | 3.99 | 96.4 | 17.7 | 76.4 | 3.74 | 98.5 | 6.62 | 93.2 |
| Synthetic only | Decoupled (MLP) | 4.19 | 97.8 | 5.04 | 96.6 | 4.40 | 97.6 | 3.73 | 97.7 | 1.27 | 100 | 15.6 | 78.9 | 11.1 | 86.8 | 4.50 | 95.7 | 19.1 | 75.0 | 3.58 | 98.8 | 7.25 | 92.5 |
| Raw real | Decoupled (MLP) | 3.48 | 98.4 | 4.29 | 97.3 | 3.75 | 98.3 | 3.37 | 98.0 | 1.32 | 100 | 15.2 | 80.1 | 8.95 | 90.2 | 3.91 | 96.7 | 18.6 | 75.7 | 3.48 | 98.4 | 6.63 | 93.3 |
| Improved real | Decoupled (MLP) | 3.53 | 98.3 | 4.30 | 97.3 | 3.69 | 98.3 | 3.16 | 98.3 | 1.33 | 100 | 15.3 | 79.3 | 9.28 | 90.1 | 3.95 | 96.6 | 17.6 | 76.8 | 3.19 | 99.0 | 6.53 | 93.4 |
| **Affine-invariant disparity** | | | | | | | | | | | | | | | | | | | | | | | |
| Improved real | Entangled (SI-Log) | 4.69 | 98.1 | 4.99 | 97.2 | 5.73 | 96.7 | 4.63 | 97.8 | 2.14 | 100 | 21.9 | 71.1 | 11.3 | 88.0 | 5.69 | 95.8 | 23.8 | 68.4 | 5.46 | 98.3 | 9.03 | 91.1 |
| Improved real | Entangled (Shift inv.) | 4.03 | 98.3 | 4.39 | 97.7 | 4.48 | 97.8 | 3.85 | 98.2 | 1.47 | 100 | 20.2 | 73.8 | 9.84 | 89.8 | 4.75 | 96.5 | 21.3 | 69.5 | 3.74 | 99.0 | 7.80 | 92.1 |
| Improved real | Decoupled (Conv) | 3.96 | 98.4 | 4.49 | 97.5 | 4.29 | 97.9 | 3.75 | 98.4 | 1.39 | 100 | 19.9 | 74.4 | 9.89 | 89.7 | 4.75 | 96.3 | 21.1 | 70.0 | 3.81 | 98.8 | 7.73 | 92.1 |
| Synthetic only | Decoupled (MLP) | 4.75 | 97.8 | 5.17 | 96.7 | 4.86 | 97.6 | 4.29 | 98.0 | 1.31 | 100 | 20.8 | 74.0 | 11.5 | 87.3 | 5.21 | 95.9 | 22.2 | 69.7 | 3.66 | 99.0 | 8.38 | 91.6 |
| Raw real | Decoupled (MLP) | 3.92 | 98.4 | 4.50 | 97.4 | 4.23 | 98.0 | 3.95 | 98.2 | 1.35 | 100 | 19.7 | 74.8 | 9.22 | 90.6 | 4.78 | 96.3 | 21.4 | 69.6 | 3.61 | 98.7 | 7.69 | 92.2 |
| Improved real | Decoupled (MLP) | 4.03 | 98.3 | 4.45 | 97.5 | 4.11 | 98.1 | 3.74 | 98.2 | 1.37 | 100 | 19.8 | 75.4 | 9.71 | 90.1 | 4.79 | 96.2 | 20.0 | 72.5 | 3.30 | 99.3 | 7.53 | 92.6 |

Table B.5: Evaluation results of ablation study on each sets

