# OpenReview forum: "MoGe-2: Accurate Monocular Geometry with Metric Scale and Sharp Details"
_NeurIPS.cc/2025/Conference — NeurIPS 2025 poster_

### Official Review · Reviewer_bgf3 · 2025-06-26

**Clarity:** 3
**Significance:** 3
**Originality:** 3
**Rating:** 4
**Confidence:** 3

**Summary:**

In this paper, MoGe-2 is proposed to extend MoGe for predicting metric geometry. The main contribution of this paper is introducing an additional branch for scale prediction, in which the global information in tokens from DINOv2 is utilized to predict an accurate metric scale. Additionally, a real data refinement approach is proposed for accurate geometry estimation. The visualization in Figure 5 shows the effectiveness of the proposed mismatch filtering scheme. The experiment results show the better performance of MoGe-2 compared to the latest ones.

**Questions:**

Please refer to the weakness part.

**Ethical Concerns:**

["NO or VERY MINOR ethics concerns only"]

**Final Justification:**

I appreciate the authors’ thorough responses, which have addressed all of my initial concerns. I recommend incorporating these discussions into the final version for completeness. Accordingly, I will retain my original positive rating.

**Limitations:**

Yes.

**Paper Formatting Concerns:**

No.

**Quality:**

3

**Strengths And Weaknesses:**

Strengths:
1. The paper is well-organized and easy to follow. The motivation of this paper is reasonable.

2. Decouple relative geometry and metric scale predictions is a nice idea, and the explanation of the reason for the final configuration is detailed.

3. The method proposed to process the raw real data is straightforward and effective, which has been shown in Figure 4 and Figure 5. The visualization clearly explains the real data refinement.

4. The qualitative and quantitative results show the effectiveness of the proposed method compared to the latest methods.


Weaknesses:
1. It is not clear how to choose the value of $\hat r_j$ in equations (4)  and (5), which should be an important parameter in using mismatch filtering.

2. I am curious about how to get the global shift $t$ in MoGe-2, since the global shift $t$ can be determined by a robust and optimal (ROE) alignment solver in MoGe, which only outputs point maps with unknown scales.

3. It would be better if the authors could explain more about 'the absolute bias of the former' in the Lines 198-199.

4. It would be better if the authors could show some failure cases with visualization, which will give an idea for further research works.

---

> ### Author Rebuttal · Authors · 2025-07-30
>
> We appreciate the reviewer’s valuable feedback and insightful questions.
>
> 1. **Regarding the choice of radius $r$ in mismatch filtering.**
>
>    Our strategy follows the same multi-scale radius selection as in MoGe's local loss, which has been verified as effective for leveraging context at different spatial scales. Specifically, we select three radii such that their projected image regions approximately cover **1/4, 1/16, and 1/64 of the image size**, respectively. This multi-scale design allows the filtering to capture and suppress misalignments of varying scales using both coarse and fine local evidence.
>     We will include this explanation and the exact computation formula in the revised version for completeness.
>
> 2. **Clarification on global shift in training.**
>
>     We assume the reviewer is referring to the *training-time* global shift used in supervision. As stated in Line 154–155, *"we maintain the geometry branch for affine-invariant point map as in MoGe, ..."* We are happy to elaborate further for clarity.
>
>     In both the main model and all scale-decoupled ablations, the affine point map branch follows **exactly the same ROE alignment strategy as MoGe** in training supervision. Specifically, the predicted affine point map is aligned to the ground truth point map via ROE alignment, which jointly solves for a global scale and shift to minimize the ROE loss (i.e., L1 under best affine alignment).
>
>     $$
>     (s^*, \mathbf{t}^\*) = \arg\min_{s, \mathbf{t}} \sum_{i \in \mathcal{M}} w_i \Vert s \hat{\mathbf{p}}_i + \mathbf{t} - \mathbf{p}_i\Vert_1
>     $$
>
>     This alignment yields:
>
>     - the optimal global scale $s^*$, which is also used as ground-truth for supervising the metric scale prediction branch as decribed in Eq.(3).
>
>     - the optimal global shift $\mathbf t^*$.
>
>     The metric scale prediction branch thus learns to regress the global scale $\hat s$ directly, with supervision from the ROE-derived $s^*$ as decribed in Eq.(3).
>
>     Only in the special ablation, *Entangled Shift-invariant*, where the predicted point map is not affine-invariant but metric-scaled with unknown shift, the alignment is done with shift only:
>
>     $$
>     \mathbf{t}^* = \arg\min_{\mathbf{t}} \sum_{i \in \mathcal{M}} w_i \Vert\hat{\mathbf{p}}_i + \mathbf{t} - \mathbf{p}_i\Vert_1
>     $$
>
>     We will make this distinction more explicit in the final version and include the relevant formulation for completeness.
>
> 3. **Clarification on "absolute bias".**
>
>     We appreciate the reviewer’s attention to this point. The term *absolute bias* refers to the systematic deviation of geometry predictions caused by the lack of real-scene prior knowledge when training purely on synthetic data. While synthetic-trained models can often recover local relative geometry accurately, they may produce biased estimation on scene-level geometry and layout when applied to real-world images.
>
>     This is especially evident in large-scale indoor or outdoor scenes, where synthetic models are unfamiliar with real-scene object proportions or layouts. As illustrated in **Figure 5**, the model trained on synthetic data reconstructs the relative shape of the chair well, but places it **too close and too small** compared to the ground truth, revealing a global misinterpretation of the scene layout. We use the term *absolute bias* to describe such behavior, where the **local structure appears plausible**, but the **global layout is misaligned**.
>
>     This observation motivates our choice of a **local filtering strategy** (rather than relying on global consistency), which is more robust to these domain-induced absolute errors. We will clarify this term further in the revision to avoid ambiguity.
>
> 4. **Failure cases**
>
>     We agree that visualizing failure cases can provide useful insights. As noted in our paper, failure typically arises in scenes with extremely fine structures, significant foreground-background scale disparity, or strong out-of-distribution ambiguity in metric scale. We will add some visualizations of the failure cases in the revised version of our paper.
>
> ---
>
> Please let us know if you have further questions.

---

> > ### Comment · Reviewer_bgf3 · 2025-08-04
> > **I will retain my original positive rating**
> >
> > I appreciate the authors’ thorough responses, which have addressed all of my initial concerns. I recommend incorporating these discussions into the final version for completeness. Accordingly, I will retain my original positive rating.

---

> > > ### Author Response · Authors · 2025-08-06
> > >
> > > Dear reviewer,
> > >
> > > Thank you for your feedback. We're glad to see that all of your initial concerns have been addressed. We will ensure that these discussions are properly incorporated into the revised paper, and we sincerely appreciate your thoughtful questions. Please feel free to let us know if there's any clarifications we shall further provide that could assist with the final assessment.
> > >
> > > Authors

---

### Official Review · Reviewer_2tk2 · 2025-06-29

**Clarity:** 3
**Significance:** 3
**Originality:** 3
**Rating:** 4
**Confidence:** 4

**Summary:**

MoGe2 extends the prior work MoGe to recover metric-scale 3D point maps from a single image while preserving local shape details. Two scale-prediction strategies (1) embedding scale into a shift-invariant point map, and (2) decouple and predict a global scale with affine-invariant representation. To sharpen fine structures, MoGe2 introduces a data-refinement pipeline that first filters out noisy real-world depth via local alignment against synthetic finetuned MoGe,  and then completes missing regions with Poisson inpainting guided by synthetic pseudo-labels. Experiments on different benchmarks demonstrate SoTA performance in relative and metric error metrics and boundary F1 scores.

**Questions:**

1. **Why MLP > Conv head?**  It is still unclear to me why CLS token prediction works better than conv head.  The authors attribute the CLS-token MLP’s superior performance to “global information”. Could you simply concatenate global DINOv2 features into the conv head to achieve the same effect?

2. **Use GT scale**? Instead of computing $\hat s^{\ast}$ by aligning the predicted point map to the GT with the ROE solver, could you simply derive a scale target directly from the GT pointmap, e.g., using the median or mean norm of the GT points as supervision? What are the pros and cons of this simpler “closed-form” approach compared to your alignment-based $\hat s^{\ast}$?

3. **Depth filtering / completion**. What happens when large portions of the depth map are filtered out? In cases of heavy misalignment (e.g., reflections), does the subsequent Poisson completion introduce noticeable bias or smoothing artifacts in the reconstructed geometry?

**Ethical Concerns:**

["NO or VERY MINOR ethics concerns only"]

**Final Justification:**

Thanks author for the detailed replies and additional experiments. I think my concerns are fully solved. I'll keep my ratings.

**Limitations:**

Yes

**Quality:**

4

**Strengths And Weaknesses:**

- Strengths

	1.	**Motivation is well stated**. The need for metric scale and fine-detail recovery in real-world applications is clearly motivated.
	2.	**Effective scale decoupling**. A lightweight CLS-token MLP head for scale prediction is more effective compare to the convolutional head prediction.
	3.	**Effective real-data filtering**. Training with refined real data with local alignment filtering and synthetic-guided inpainting, yields better geometry without sacrificing accuracy, demonstrating that incorporating real data rather than only synthetic is key to fine-detail recovery
	4.	**Comprehensive evaluation achieves SoTA**. Across multiple diverse datasets, MoGe2 sets new SoTA in both relative-geometry and metric-depth benchmarks.
	5.	**Simplicity**. The proposed pipeline builds on MoGe’s framework  robust alignment solver and multi-scale L1 loss with minimal extra supervision or complex architectural changes.

- Weaknesses
	1.	**Limited novelty**. Key contribution (additional scale prediction and real data filtering / completion) are straightforward extensions of MoGe; It feels more like a solid engineering effort that carefully stitching together existing techniques and finetuning them.

	2.	**Supervision for metric scale is under-explained**. Equation (3) uses the alignment-derived scale $\hat s^{\ast}$ from MoGe as a training target. This supervisory signal is only mentioned in passing on lines 124–125. The paper doesn't detail how $\hat s^{\ast}$ is computed via the ROE solver. A detail description of the optimal-scale calculation and the justification is needed.

---

> ### Author Rebuttal · Authors · 2025-07-30
>
> We thank the reviewer for the detailed and constructive feedback, as well as the recognition of our motivation, design, and experimental results. We reply to the reviewer's concerns as follows.
>
> 1. **Concerns about Novelty.**
>
>     We would like to clarify the novelty and key insights behind our design:
>     - **Network design for scale prediction:** While the final design of using a CLS-token MLP appears simple, it is the result of both **a key insight and careful empirical validation**.
>     This insight is that **metric scale is inherently a global property** and is better predicted by decoupling from affine dense prediction and leveraging global feature.
>     Importantly, this was **not an arbitrary architectural choice**. We experimentally compared several alternative designs and found that the CLS-token MLP yielded better results (Tab. 4).
>     To the best of our knowledge, this specific strategy has not been explored or experimentally validated in prior works on metric depth or 3D point map recovery.
>     - **Real-world data refinement**: To achieve ***both*** real-world geometry ***accuracy*** and ***sharp*** details, we propose a nontrivial refinement pipeline that addresses structured artifacts commonly present in real-world RGB-D data (e.g., SfM, LiDAR). Our empirical observations reveal a duality in such labels that they are partially accurate but often degraded by systematic noise. The key challenge and also motivation behind our design is to ***exploit the reliable regions while avoiding degradation from corrupted areas***. Our method selectively filters out noisy regions and fills them in using a model trained on clean synthetic data. This pipeline is essential for bridging the synthetic-to-real domain gap, enabling robust and high-fidelity supervision across diverse real-world sources.
>
>     In summary, our designs emphasize simplicity without compromising effectiveness. We believe this type of simple design with strong performance, supported by analysis and experiment, constitutes a meaningful contribution.
>
> 2. **MLP v.s. Conv head.**
>
>     We thank the reviewer for the insightful suggestion. The proposed alternative, which repeats and concatenates the CLS-token feature to the Conv head input for metric scale prediction, indeed introduces global information and partially decouples the scale from structure.  We implemented the reviewer's suggested design and found that **it does improve performance over Conv head without CLS-token feature**, confirming that global features are important for predicting scale. However, **it remains less effective than directly applying an MLP to the CLS-token**. As summarized in the table below, the MLP head demonstrates slightly better overall performance (*wins in 11 out of 17 metrics, ties in 2*), and remains the most straightforward and effective design while being simple and lightweight in practice.
>
>
>     | Configuration | \| | Metric |  |  | \| |   |   |   |   | Relative |   |  |  |  ||  | \|Sharpness      |
>     | ------------- | --- | -----| -----|-- | ----- | -----|----- | -----|--|----|------ | ------ | ----- | ----- | ------|-----|------- |
>     |               | \|Point           |       | \|Depth     ||\|Point |  |    |     |      | |\|Depth |||||  |\|
>     | | | | | | \|Scale-inv. | | Affine-inv. | | Local |  | \|Scale-inv. || Affine-inv. || Affine-inv.(disp)| |\| |
>     |  | \|Rel↓ | δ₁↑ | Rel↓ | δ₁↑ | \|Rel↓ | δ₁↑ | Rel↓ | δ₁↑ | Rel↓ | δ₁↑ | \|Rel↓ | δ₁↑ | Rel↓ | δ₁↑ | Rel↓ | δ₁↑ | \|F1   |
>     | Conv head | \|9.62 | 91.4 | 17.6 | 68.4 | \|12.2 | 86.3 | 9.15 | 90.0 | 6.34 | 94.9 | \|8.46 | 90.2 | 6.62 |93.2 | 7.74 | 92.1 | \|**12.7** |
>     | CLS token + Conv head  | \|**9.02** | **92.0** | 16.7 | 69.7 | \|11.9 | **87.6** | 8.93 | 90.5 | **6.21** | **95.1** | \|8.36 | 90.6 | 6.57 | **93.5** | 7.62 | 92.3 | \|12.3 |
>     | CLS token + MLP head  | \|9.20 | 91.9 | **16.5** | **72.8** | \|**11.6** | **87.6** | **8.87** | **90.6** | 6.26 | **95.1** | \|**8.23** | **91.0** | **6.53** | 93.4 | **7.53** | **92.6** | \|12.5 |
>
>
> 3. **Using of GT scale factor ($s^*$).**
>
>     We thank the reviewer for pointing out that the details of the ground-truth scale $s^\*$ were under-explained. As proposed in MoGe, the $s^*$ is the optimal scale solution under a scale-and-shift alignment with respect to a weighted *least absolute deviation* (LAD) problem:
>     $$
>     (s^\*, \mathbf{t}^\*) = \arg\min_{s, \mathbf{t}} \sum_{i \in \mathcal{M}} w_i \Vert s \hat{\mathbf{p}}_i + \mathbf{t} - \mathbf{p}_i\Vert_1
>     $$
>     The ROE solver divides this problem into a series of searching subproblems in parallel, utilizing the piece-wisely linear property of the objective landscape. We will include additional explanation to supplement the preliminary section.
>
>     Regarding the **choice of GT scale**, we believe the use of the ROE-derived scale $s^\*$ as ground truth is well-motivated and principled. By definition, $s^\*$ **minimizes exact ROE loss to best match $s \hat{\mathbf{P}}$ with $\mathbf{P}$**, and is thus a natural supervisory signal aligning with the training strategy of the affine-invariant ROE loss in MoGe  to inherit its superior performance.
>
>     To address the reviewer’s suggestion of using **a simpler closed-form target (e.g., mean distance from centroid)**, we implemented a variant using median-centered normalization and scale defined by average distance—similar to the approach in MiDaS. However, this alternative led to **degraded performance** on metric prediction:
>     - Metric depth AbsRel error increased from **16.5% → 17.8%**,
>     - Metric point AbsRel error increased from **9.20% → 10.2%**.
>
>     These results confirm that using ROE-aligned $s^*$ yields better supervision under our task formulation.
>
> 4. **Depth filtering / completion on large missing areas.**
>
>     Our refinement pipeline remains effective even when large regions are filtered out. Although the inpainted content may not be perfectly accurate, the **Poisson inpainting preserves structural sharpness and edge alignment due to its gradient-based formulation**. Crucially, the filled regions are at worst comparable in quality to the original synthetic predictions, ensuring that supervision quality is not degraded. In contrast, leaving these regions empty would create large unsupervised areas during training, which could harm model performance in those regions.
>
> ---
> Please let us know if you have any further questions.

---

> > ### Comment · Reviewer_2tk2 · 2025-08-04
> > **The response solve my concerns**
> >
> > Thanks author for the detailed replies and additional experiments. I think my concerns are fully solved. I'll keep my ratings for now if no other comments from other reviewers.

---

> > > ### Author Response · Authors · 2025-08-06
> > >
> > > Dear reviewer,
> > >
> > > Thank you for your comment. We're glad to see that your concerns have been fully solved. Please feel free to let us know if there's any clarification we shall further provide that could assist with the final assessment. Thanks again.
> > >
> > > Authors

---

### Official Review · Reviewer_vmnM · 2025-07-01

**Clarity:** 3
**Significance:** 2
**Originality:** 2
**Rating:** 3
**Confidence:** 5

**Summary:**

This paper focuses on improving the geometric accuracy, metric scale estimation, and geometric detail quality for monocular metric point cloud reconstruction from a single image. The authors propose an extension of the previous MoGe method to jointly estimate an affine-invariant point cloud along with a global metric scale. The metric scale is predicted by extracting image-level information stored in the CLS token through an additional MLP head. To better utilize real-world data, the authors filter out inaccurate ground truth regions and complete them using a MoGe model trained solely on synthetic data.

**Questions:**

See above

**Ethical Concerns:**

["NO or VERY MINOR ethics concerns only"]

**Limitations:**

yes

**Quality:**

2

**Strengths And Weaknesses:**

Strengths contributions:
1. The paper is clearly written and easy to follow, which helps in understanding both the technical contributions and experimental results.
2. The authors provide extensive quantitative evaluations, including relative and metric geometric accuracy and F1-score sharpness, demonstrating the robustness and reliability of the proposed approach.
3. The proposed robust, accurate, and fine-grained monocular point cloud estimation method has practical value and can benefit the research community in relevant areas.




Limitations weaknesses:


My primary concerns relate to the novelty of the proposed contributions. While the method is robust and practically valuable, its conceptual novelty appears somewhat limited.


Major Concerns:

1. Novelty of Metric Scale Estimation.

The design for metric scale estimation shown in Figure 3(a) shares similarities with existing approaches such as ZoeDepth[1] and UniDepth[2]. Moreover, for Figure 3(c), the use of the CLS token to capture global image-level information for scale estimation is conceptually aligned with prior works like ScaleDepth[3] and RSA[4], which also predict global scale based on features from CLIP or language models. The main contribution of MoGe-2 appears to be extending metric scale estimation from depth estimation to point cloud estimation frameworks.


2. Novelty of Leveraging Real-World Data.

The proposed approach filters inaccurate regions in real-world ground truth and completes them using a model trained on synthetic data. This design is similar to the training strategy of DepthAnything-v2, which initially trains on synthetic data and further refines on pseudo-labels from real data. The main difference lies in using RGB-D rather than RGB data for pseudo-label generation. However, compared to the diversity of RGB data, the advantages of RGB-D remain unclear. An ablation study of "w/ improved real data" and "w/ pseudo label of the same RGB real data", would help clarify its effectiveness better.

Minor Comments:

Line 178, "However, this still limits geometry accuracy because synthetic data rarely captures real-world diversity." While this highlights the motivation for incorporating real-world data, to my knowledge, the diversity and quality of synthetic datasets are not inherently inferior to those of real-world RGB-D datasets. Therefore, this justification may not be entirely convincing from my perspective.


[1] Bhat, Shariq Farooq, et al. "Zoedepth: Zero-shot transfer by combining relative and metric depth." arXiv preprint arXiv:2302.12288 (2023).
[2] Piccinelli, Luigi, et al. "UniDepth: Universal monocular metric depth estimation." Proceedings of the IEEE/CVF Conference on Computer Vision and Pattern Recognition. 2024.
[3] Zhu, Ruijie, et al. "Scaledepth: Decomposing metric depth estimation into scale prediction and relative depth estimation." arXiv preprint arXiv:2407.08187 (2024).
[4] Zeng, Ziyao, et al. "RSA: Resolving Scale Ambiguities in Monocular Depth Estimators through Language Descriptions." The Thirty-eighth Annual Conference on Neural Information Processing Systems.

---

> ### Author Rebuttal · Authors · 2025-07-30
>
> ### On the novelty of scale prediction.
> ---
>
> We appreciate the reviewer’s feedback and the insightful references.
>
> #### **Clarification about the references**
>
> 1. > *"The design for metric scale estimation shown in Figure 3(a) shares similarities with existing approaches such as ZoeDepth[1] and UniDepth[2]"*
>
>     *Figure 3 has three subfigures, 3-1, 3-2(a) and 3-2(b). We assume the reviewer is referring to either* ***3-1*** *or* ***3-2(a)*** *(please let us know otherwise)*
>
>     We would like to clarify that both **Figure 3-1** ("Entangled") and **Figure 3-2(a)** ("Decoupled, Conv head"), which the reviewer associates with ZoeDepth and UniDepth, are in fact part of our **ablation baselines** for the purpose of validating our **final design in Figure 3-2(b)**.
>
>     While these baseline settings may resemble prior works, they are **not the MoGe-2 architecture**, but part of our contribution to provide controlled comparisons and demonstrate the effectiveness of our decoupled metric scale estimation design.
>
>     - **UniDepth** separates camera and depth modules, **but does not decouple global metric scale from depth**. In fact, our ablation experiment **"Entangled + SI-log Loss"**  is designed to replicate UniDepth's formulation and learning objectives (SI-log for depth, MSE for camera UV map) to offer a fair and controlled baseline comparison.
>
>     - **ZoeDepth** adopts an **AdaBins-style** formulation with **per-pixel bin classification** for both depth and metric modules.  While ZoeDepth includes a scale estimation component, its formulation and objective are different from our **simpler direct regression design**.
>
> 2. > *"for Figure 3(c), the use of the CLS token to capture global image-level information for scale estimation is conceptually aligned with prior works like ScaleDepth[3] and RSA[4]"*
>
>     *We assume the reviewer is referring "Figure 3(c)" to the 3rd subfigure, Figure 3-2(b).*
>
>     Regarding **ScaleDepth** and **RSA**, while we acknowledge their use of global features to resolve metric scale, they both **rely on text input** and **language models** (e.g. CLIP) to provide such semantic feature. In contrast, our method explores how to leverage **purely image-based features** from **visual-only pretraining** (e.g., DINOv2), tailored for metric point map estimation. Our approach tackles a different and much simpler setting, and is also evaluated as effective by experiments.
>
> #### **Clarification of the novelty of our scale prediction**
>
> To our knowledge, no prior work has proposed or systematically validated directly predicting the global scale from CLS tokens in this context, with exclusive supervision and loss alignment as we propose.
> Behind the simple yet effective final design is the result of both a key insight of decoupling relative geometry and metric scale and careful empirical validations. We believe this type of simple design with strong performance, supported by analysis and experiment, constitutes a meaningful contribution.
>
> &nbsp;
>
> ### On novelty of leveraging real-world RGB-D data
> ---
>
> #### **Differences from Depth Anything V2**
>
> > *" This design is similar to the training strategy of DepthAnything-v2... The main difference lies in using RGB-D rather than RGB data for pseudo-label generation."*
>
> We wish to point out that the **goals** and **mechanisms** of Depth Anything V2's pseudo labeling and our real-world data pipeline **differ fundamentally**. They serve **orthogonal purposes** as summarized below:
>
> | Method                | Synthetic Data (e.g. Hypersim) | Real-World Labeled Data (e.g. Taskonomy) | Real-World Unlabeled Data (e.g. OpenImageV7) | Purpose                                   |
> | --------------------- | ------------------- | ------------------------------------ | ------------------------------ | ----------------------------------------- |
> | **Depth Anything V2** | √ Used | × Not used   | √ Pseudo-labeled                | Transfer knowledge from a *stronger teacher model* via pseudo labels on RGB-only data. |
> | **Ours**              | √ Used | √ Refined    | × Not used                              | Learning accurate depth supervision from *real-world RGB-D* while mitigating sensor and structural artifacts  |
>
> Depth Anything v2 focuses on distillation using the pseudo labels generated by a stronger teacher model, where the *unlabeled* *diverse* RGB images serve as a bridge for knowledge transfer. In contrast, we do incorporate real RGB-D ground truth labels, and propose a new filtering and refinement pipeline to address its artifacts to **improve metric geometry accuracy without sacrificing sharpness**.
>
> Using a larger teacher model for pseudo-label distillation is **orthogonal** to our real-world data refinement pipeline. In fact, the two strategies could be used **on top of each other**, e.g., further distilling from a ViT-Giant MoGe-2 trained on refined real data.
>
> To isolate the impact of pseudo-labeling itself—rather than gains from scaling model size, we followed the reviewer’s suggestion and **replaced our filtered real GT depth with pseudo-labels** from a model of **the same capacity** trained with synthetic data in ablation studies (ViT-B). As shown in the table below, the resulting performance is close to using synthetic-only training, and significantly inferior than using real labels.
>
> | Configuration | Metric Point Rel↓ | Metric Point δ₁↑ | Metric Depth Rel↓ | Metric Depth δ₁↑ |
> | ------------- | --- | -----| -----|--- |
> Synthetic data only| 12.4 | 87.3 | 21.7 | 65.0 |
> w/ Pseudo label| 12.5 | 87.4 | 21.6 | 65.8 |
> w/ Improved real label| **9.20** | **91.9** | **16.5** | **72.8**
>
> These results indicate that **pseudo labels alone do not provide free gains from a same-capacity model**, and highlight the value of leveraging refined real RGB-D label for metric supervision.
>
> #### **The necessity and effectiveness of real-world RGB-D data**
>
> > *"The diversity and quality of synthetic datasets are not inherently inferior to those of real-world RGB-D datasets"*
>
> We respectfully disagree with the reviewer's assertion that (currently available) synthetic data alone can sufficiently cover the real-world RGB-D distributions.
>
> As clearly shown in **Table 4** in the paper, compared to those trained with real data, models trained solely on synthetic data suffer from a **notable drop in accuracy** (e.g. metric depth AbsRel increases 15.8%→21.7%, affine-invariant disparity 7.53%→8.37%) on all metrics except for sharpness.
> This reflects the **known limitations of synthetic data**: *Domain gap* and *limited scene coverage* (also recognized in Depth Anything V2's Section 3, but they choose to use giant model distillation to improve large and base model's performance).
>
> Our work builds on this understanding, and provides a principled and effective pipeline to **incorporate real RGB-D data** while **mitigating its artifacts to avoid degrading sharpness**. This is a core motivation backed by both existing literature and our ablation results.
>
> ---
>
> Please let us know if you have further questions.

---

> > ### Author Response · Authors · 2025-08-06
> >
> > Dear reviewer,
> >
> > Thank you again for your valuable feedback. As the discussion phase nears its end, we wanted to kindly bring your attention to our rebuttal and would be glad to provide any further clarification if needed.
> >
> > Authors

---

> > > ### Comment · Area_Chair_SvuY · 2025-08-07
> > >
> > > Dear vmnM,
> > >
> > > We'd love to hear your thoughts on the rebuttal.
> > > If the authors have resolved your (rebuttal) questions, please do tell them so. If the authors have not resolved your (rebuttal) questions, please do tell them so too.
> > >
> > > Thanks,
> > > Your AC

---

> > > > ### Author Response · Authors · 2025-08-09
> > > > **Summary of reviewer vmnM’s review and our clarifications**
> > > >
> > > > Dear AC and all reviewers,
> > > >
> > > > As the discussion phase will end in less than one hour and Reviewer vmnM has not responded, we’d like to briefly summarize the reviewer's major concerns and our responses for your reference.
> > > >
> > > > - The reviewer raised concerns about the novelty of our metric scale prediction design compared to some prior works. We clarified that: 1) the reviewer has misinterpreted our figure and associated ZoeDepth and UniDepth to our ablated baseline design, not our final model, and 2) ScaleDepth and RSA require language input and their use of global features is from the text-CLIP model, which clearly differs from ours that exploits purely visual global feature from CLS token of the ViT model.
> > > > - The reviewer claimed similarity between our real-data refinement pipeline and that of Depth Anything V2. We clarified that the two approaches are fundamentally different in both mechanism and purpose: ours refines real RGB-D labels for accurate metric supervision, while Depth Anything V2 focuses on pseudo-label distillation from larger models.
> > > > - The reviewer questioned the necessity of real-world RGB-D data over synthetic data. We respectfully disagree and pointed out strong empirical evidence in Table 4, as well as supporting references from prior work.
> > > >
> > > > We thank all reviewers again for their constructive feedback. We hope the above summary can assist with your final assessment in the absence of Reviewer vmnM’s interaction with us.
> > > >
> > > > Authors

---

### Official Review · Reviewer_RFUA · 2025-07-06

**Clarity:** 4
**Significance:** 4
**Originality:** 4
**Rating:** 5
**Confidence:** 3

**Summary:**

## Paper Summary
This paper addresses the challenging task of monocular geometry prediction through two key contributions:
- Decoupled Geometry and Scale Prediction:
    - Geometry prediction follows a method similar to prior work (MoGe).
    - Scale is predicted using an MLP that processes the DINOv2 CLS token.
- Novel Data Pipeline for Real-World Training:
    - A two-step preprocessing approach improves training on real data:
        - Removal of incorrect depth estimates.
        - Inpainting of missing depth values.

The method is evaluated on multiple standard benchmarks, demonstrating significant improvements over previous state-of-the-art approaches.

## Review Summary
The technical contributions are well-motivated and sounding, supported by comprehensive experiments. I don't see any major weaknesses.

**Questions:**

1. Was the DINOv2 backbone fine-tuned or kept frozen during scale prediction training?
If frozen, what is the hypothesized relationship between the CLS token and scene scale? For instance: Could a correlation analysis (e.g., x-axis: object/scene categories, y-axis: predicted scales) reveal biases (e.g., indoor, outdoor, and object-centric)? Does the MLP implicitly learn to classify scene types (e.g., "object-centric" vs. "wide-area") to infer scale?

2. In Fig. 6 (3rd example), testing on a generic building may not fully validate scale accuracy. Maybe use landmarks with known heights (e.g., Eiffel Tower, Burj Khalifa, Taipei 101) to quantitatively compare predicted vs. true scales across methods. This could better demonstrate whether the decoupled scale predictor generalizes to extreme but measurable cases.

**Ethical Concerns:**

["NO or VERY MINOR ethics concerns only"]

**Final Justification:**

I appreciate the detailed author response and I don't have further questions. I'd like to keep my original positive score.

**Limitations:**

Yes

**Quality:**

4

**Strengths And Weaknesses:**

## Strengths
- The paper is very well written and easy to follow with strong analysis and visualisations.
- The idea of decoupling scale prediction from geometry is simple and effective. It's interesting to see DINOv2 cls token can be used for scale prediction.
- The proposed two-step data pipeline (incorrect depth removal + inpainting) effectively enables training on real-world data while mitigating inaccuracies in depth measurements.
- Experiments are comprehensive, covering multiple benchmarks, and results show promising improvements over prior work.

## Weaknesses
I don't see any major weaknesses. I do have a few questions (see Questions section).

---

> ### Author Rebuttal · Authors · 2025-07-30
>
> We sincerely thank the reviewer for the constructive and encouraging feedback. Your clear summary and thoughtful questions help us better articulate the implications and potential extensions of our work.
>
> 1. **DINOv2 backbone finetuning; Scale correlation with scene types.**
>
>     All model parameters, including the DINOv2 backbone, are trainable during training. We follow the same initialization and optimizer setup as MoGe: DINOv2 pretrained weights for the backbone, and randomly initialized heads, trained end-to-end (see lines 219–225 in Implementation Details).
>
>     The reviewer raises an insightful point regarding the potential correlation between the CLS token and scene types. Indeed, the CLS token of DINOv2 encodes strong global semantics. For example, according to DINOv2's official benchmarks, the frozen ViT-Large backbone's CLS token achieves **83.5% (k-NN)** and **86.3% (linear probe)** top-1 accuracy on ImageNet classification, demonstrating its strong capacity to represent scene and object semantics. We believe this capacity to implicitly recognize scene types and iconic objects with known sizes is directly relevant to metric scale estimation.
>
>     As suggested by the reviewer, we analyzed the predicted global scale factor from the MLP across two datasets:
>
>     - **Indoor (NYUv2)**: Mean = 1.79, Std = 0.44
>
>     - **Street-view (KITTI)**: Mean = 11.9, Std = 2.0
>
>     This clustering suggests that our MLP head over the CLS token indeed captures meaningful scene-type-aware priors for metric scale estimation.
>
> 2. **Including more examples of recognizable landmarks**
>
>     We appreciate the reviewer’s thoughtful suggestion. While the rebuttal format limits us from including more examples, we clarify that the scene in **Figure 6 (3rd row)** comes from a real-world photo of **135 East 57th Street (Tower Fifty Seven)** in Manhattan, New York. This building has a known height of **131.1 m**, and our model estimates it as **116 m**. In the same image, a **New York yellow taxi** (**Ford Crown Victoria**, approximately **1.45** m tall) is also present. Our model predicts it as **1.49 m**. These consistent predictions across both large-scale and small-scale objects suggest that our model can effectively ground metric estimates using recognizable reference cues in the scene.
>
>     We agree that using landmarks with known heights is a valuable way to validate scale estimation accuracy. At the same time, we note that certain landmarks with **highly distinctive and atypical structures** may present challenges for our model, due to the lack of familiar contextual cues. For example, when tested on images of the **Eiffel Tower**, the model estimates a height of **70–100 m**, which may appear reasonable in isolation but underestimates the **true height (330 m)**. Interestingly, on the Las Vegas replica, which shares a similar appearance but stands at roughly half the height (**165 m**), the model predicts a similar range of **70–120 m**, which is closer to the true height. These cases highlight potential areas for future improvement and underscore the importance of visual context in scale reasoning. We will release an open demo to help users explore and better understand such generalization behaviors.
>
> ---
>
> Please let us know if you have further questions.

---

> > ### Comment · Reviewer_RFUA · 2025-08-05
> >
> > I appreciate the detailed author response and I don't have further questions. I'd like to keep my original positive score.

---

> > > ### Author Response · Authors · 2025-08-06
> > >
> > > Thank you for your comments. We're glad to see that our response has addresses your questions. Our paper will be revised accordingly. Thanks again.
> > >
> > > Authors

---

### Decision · Program_Chairs · 2025-09-17

**Decision:**

Accept (poster)

**Comment:**

The paper extends MoGe for zero-shot metric depth estimation by predicting absolute scale from the CLS-token of the underlying DINOv2 architecture, and proposes a procedure for refining real-world RGB-D datasets.

The reviewers appreciated the simplicity and effectiveness of the scale prediction, the effectiveness of the proposed filtering and refinement of real-world data, the clear writing, and resulting improvements over prior work.
Reviewers were initially concerned about the novelty of the scale prediction and the refinement procedure, and missing implementation details. The rebuttal and discussion clarified differences to prior work and effectively resolved these concerns. Reviewers recommending acceptance found their concerns resolved, while the remaining reviewer vmnM neither engaged in the discussion nor updated their final rating. The AC agrees with the assessment and recommends accepting.